# Debiasing Federated Learning with Correlated Client Participation

**Zhenyu Sun**[*], **Ziyang Zhang**[*], **Zheng Xu**[†], **Gauri Joshi**[‡], **Pranay Sharma**[‡], **Ermin Wei**[*]

[*]Northwestern University, [†]Google Research, [‡]Carnegie Mellon University

```
{zhenyusun2026,ziyangzhang2024}@u.northwestern.edu,
xuzheng@google.com, {pranaysh,gaurij}@andrew.cmu.edu,
              ermin.wei@northwestern.edu
```

## Abstract

In cross-device federated learning (FL) with millions of mobile clients, only a small subset of clients participate in training in every communication round, and Federated Averaging (FedAvg) is the most popular algorithm in practice. Existing analyses of FedAvg usually assume the participating clients are independently sampled in each round from a uniform distribution, which does not reflect real-world scenarios. This paper introduces a theoretical framework that models client participation in FL as a Markov chain to study optimization convergence when clients have non-uniform and correlated participation across rounds. We apply this framework to analyze a more practical pattern: every client must wait a minimum number of $R$ rounds (minimum separation) before re-participating. We theoretically prove and empirically observe that increasing minimum separation reduces the bias induced by intrinsic non-uniformity of client availability in cross-device FL systems. Furthermore, we develop an effective debiasing algorithm for FedAvg that provably converges to the unbiased optimal solution under arbitrary minimum separation and unknown client availability distribution.

## 1 Introduction

The massive amounts of data generated on edge devices such as cellphones or sensors offers an opportunity to train machine learning (ML) models for various applications. However, communication and privacy constraints of edge devices preclude the transfer of raw data to the cloud. Federated learning (FL) (McMahan et al., 2017; Kairouz et al., 2019; Li et al., 2020a; Yang et al., 2019) has emerged as a powerful framework to operate within these constraints by keeping decentralized data on the edge devices and instead moving model training to the edge. Federated model training operates in communication rounds. In each round, the current model is sent by the central server to edge clients, which perform model updates using their own local data, and the resulting models are then averaged by the central server. A typical cross-device FL framework consists of millions of intermittently connected edge clients, in each round only a small subset of them participate in training (Bonawitz et al., 2019). The subset of participating clients is affected by devices' intrinsic properties such as battery status and network connectivity, and also system induced constraints for efficiency and privacy. In this paper, we seek understand the effect of such client participation patterns on convergence of federated training.

The federated averaging (FedAvg) algorithm and its variants are widely used in practice (Kairouz et al., 2019; Wang et al., 2021; Hard et al., 2018; Xu et al., 2023), and the convergence has been extensively analyzed in literature (Li et al., 2020b; Woodworth et al., 2020; Wang et al., 2022; Karimireddy et al., 2019; Wang & Joshi, 2021; Wang et al., 2020). However, most works assume uniform client participation which ensures that the model update applied to the global model is an unbiased estimate of the model update in the full client participation setting. This enables convergence results for the full-participation setting to be extended to the partial participation setting resulting in an additional variance term appearing in the convergence bound (Jhunjhunwala et al., 2022; Karimireddy et al., 2019; Wang et al., 2020). A generalization of the uniform client participation model is to consider that each client has an intrinsic availability probability $p_i$ that is either known or unknown

to the central server. The set of participating clients is chosen according to this probability. Such non-uniform client participation introduces a bias in the model updates received by the server, with more frequently participating clients dominating the average update. To counter the bias, the central server can normalize the updates by the corresponding availability probabilities (Wang & Ji, 2022; Cho et al., 2022) or their estimates (Wang & Ji, 2023; Ribero et al., 2022). We consider the setting of unknown client availability and analyze the convergence.

Both the uniform and non-uniform client participation models described above assume that client participation follows a Bernoulli process that is independent across clients and rounds. This assumption fails to capture practical settings where the client participation are correlated across rounds due to memory or time-dependence constraints. In cross-device FL systems, a device can only be available for training when it is plugged in for charging, connected to unmetered network and not being actively used by the owner (Hard et al., 2018; Paulik et al., 2021; Huba et al., 2022). These criterion, which typically occurs during the night of the devices' local time, not only results in the client availability probability for non-uniform client participation, but also correlated client participation of a periodic pattern due to user preference and time zone (Kairouz et al., 2019; Eichner et al., 2019; Zhu et al., 2021). More recently, a new criteria is introduced on devices in a FL system to impose a minimum separation constraint on successive participation instances of a client (McMahan & Thakurta, 2022; Xu et al., 2023). Specifically, once a client participates in training, it cannot become available to participate for at least $R$ more rounds ($R$ specified by the central aggregating server). The minimum separation is introduced to effectively combine differential privacy (DP) and FL (Kairouz et al., 2021; Choquette-Choo et al., 2023; McMahan et al., 2024) as advanced privacy-preserving methods, and quickly becomes the default criterion in many FL applications (Xu et al., 2023; Xu & Zhang, 2024). The client participation across rounds are correlated under the minimum separation criterion, and the extreme case of very large $R$ will force cyclic client participation as studied in (Cho et al., 2023; Malinovsky et al., 2023). However, setting $R$ to be the exact value for cyclic client participation can be challenging and may cause system slowdown, and these recent work did not study non-uniform client participation or the large spectrum of minimum separation $R$ in practice. Other existing convergence analyses of federated training with generalized client participation (Wang & Ji, 2022; Rodio et al., 2023) do not fully explain the effect of such correlated client participation patterns, calling for new theoretical advances. See Appendix A for more related work discussions.

In this paper we bridge the gap of algorithms in practical FL system and the theoretical guarantees on their convergence with correlated client participation and unknown client availability. Our paper makes the following key contributions: (1) To the best of our knowledge we are the first to analyze the convergence of FedAvg with a minimum separation constraint on successive participation instances of each client, which is a general setting widely used in practical FL systems. We show that such correlated participation patterns can be captured by a Markov chain model. (2) We show that as the minimum separation $R$ increases, the effective client participation probabilities become more uniform and reduces the asymptotic bias in the solution attained by the FedAvg algorithm. (3) We propose a debiased FedAvg algorithm that estimates the unknown client participation probabilities and incorporates them in the local updates. We prove that this algorithm achieves an unbiased solution that is consistent with the global FL objective under arbitrary minimum separation $R$.

**Notations:** For any positive integer $N$, we denote $[N] = \{1, \ldots, N\}$. Let $\|\cdot\|$, $\|\cdot\|_1$ and $\|\cdot\|_\infty$ denote $l_2$-norm, $l_1$-norm and $l_\infty$-norm, respectively. For an ordered sequence $\{i_1, \ldots, i_k\}$, it is represented by $(i_1, \ldots, i_k)$ and we use the same notation for a vector when the context causes no confusion. Unless otherwise specified, $\mathbb{E}(\cdot)$ means the total expectation taken on all randomness. We use $\mathbf{c}$ to denote the vector where all entries are $c$. The $d$-dimensional Euclidean space is denoted by $\mathbb{R}^d$, and $\mathbb{R}^d_+$ is the space formed vectors where every entry is strictly positive.

## 2 PROBLEM FORMULATION

We consider the federated learning setting where $N$ clients cooperate to minimize the following global objective:

$$\min_x \quad F(x) := \frac{1}{N} \sum_{i=1}^{N} f_i(x) \tag{1}$$

where $f_i$ is the local objective function of client $i$. We aim to solve problem equation 1 in the federated learning setting, i.e., the system implements the some federated learning algorithm which

operates in rounds. In each round, a subset of the clients participate in training, and each of the clients in the subset performs multiple local updates based on the local gradients and then communicates with the server.

**Non-uniform and correlated client participation.** In this paper, we consider the scenario where each client requires some resting periods between participation and hence the participation pattern is correlated over time. Specifically, once participating in the system, an client has to wait as least $R$ rounds until its next participation, where $R$ is called *the minimum separation*. In other words, suppose client $i$'s last participation is in round $t_i$. It may join again at any round $t$ with $t \geq t_i + 1 + R$ and not before then. Moreover, when a client is available to be sampled, instead of assuming uniform sampling, we consider that each client is associated with some *unknown* strictly positive scalar $p_i > 0$ to characterize its intrinsic willingness to be sampled at every round. Without loss of generality, we assume $\sum_{i=1}^{N} p_i = 1$ and hence refer to $p_i$ as *the availability probability* of client $i$. Therefore, the client participation pattern is as follows: at each communication round, client $i$ is sampled to participate in the training process with probability proportional to $p_i$ if it has waited for $R$ rounds after its last participation; otherwise client $i$ cannot be sampled.

The above setting encompasses many of those in existing literature as special cases. For instance, note that $R = 0$ means each client is sampled at every round with probability $p_i$ independently, which is consistent with (Wang & Ji, 2023). And the cyclic participation corresponds to the case $R = \frac{N}{B} - 1$ where $B$ number of clients are sampled in each round (Cho et al., 2023), assuming the total number of clients in the FL population $N$ is divisible by $B$. We investigate the potential bias introduced by the non-uniform and correlated client participation on federated algorithm performance and propose debiasing scheme to mitigate it.

## 3 Markov Chain Model and its Properties

In this section, we propose a Markov chain model to capture the correlated participation scenario described above. Intuitively, the fact that every client cannot be sampled again within $R$ rounds motivates us to maintain a memory window with length $R$ to track which clients have not waited for $R$ rounds. In other words, clients that are possible to be sampled in the current round only depend on which clients appearing in the memory window. This calls for a Markov chain with $R$-memory, also known as $R$-order Markov chain, defined as below.

**Definition 1.** *Let $\{X_t\}_{t=0}^{\infty}$ be a stochastic process where $X_t \in \mathcal{X}, \forall t \geq 0$. It is said to be an $R$-order Markov chain if*

$$P(X_t \mid X_{t-1}, X_{t-2}, \ldots, X_0) = P(X_t \mid X_{t-1}, \ldots, X_{t-R}), \ \forall t \geq R.$$

$\mathcal{X}$ *is called the state space.*

If $R = 1$ it reduces to conventional Markov chain; if $R = 0$, then the clients can be sampled at each round with probability $p_i$, independent of the history. In a conventional Markov chain (with $R = 1$) with finite state space $\mathcal{X}$, we can use the transition probability matrix $P$ to represent the Markov chain, where the $(i, j)$-th entry of $P$ is $[P]_{i,j} = P(X_t = j \mid X_{t-1} = i)$, i.e., the probability of transitioning from state $i$ to state $j$.

Recall that each client $i$ is associated with a strictly positive availability probability $p_i > 0, \forall i \in [N]$. At each round $t$, the server samples a size-$B$ subset of clients $S_t$, where $|S_t| = B$, with probability for each client proportional to $p_i$ to join the training system. Note that only clients that have waited for $R$ rounds are available. In other words, set $S_t$ is sampled with probability proportional to $\sum_{i \in S_t} p_i$ from all subsets of size $B$ formed by the available clients. We assume $N = MB$ for some $M > 0$ and note that the minimum separation $R$ ranges from 0 to $M - 1$, where $R = M - 1$ corresponds to a cyclic participation pattern where subsets of clients participate in training in a fixed order.[1]

Denote $\mathcal{X}$ as the collection of all possible ordered subsets of $[N]$ with exactly $B$ elements. Then, $|\mathcal{X}| = \sigma(N, B)$ where $\sigma(N, B) = \frac{N!}{(N-B)!}$ represents the total number of $B$-permutations of $[N]$.

---

[1]Any $R > M - 1$ would resulting in periods with insufficient available clients. We do not consider those cases here.

Considering the stochastic process $\{X_t\}_{t=0}^{\infty}$ where $X_t \in \mathcal{X}$, the participation pattern in Section 2 can be precisely described by an $R$-order Markov chain defined in Definition 1. Formally,

$$P(X_t = \mathcal{I}_0 \mid X_{t-1} = \mathcal{I}_1, X_{t-2} = \mathcal{I}_2, \ldots, X_0 = \mathcal{I}_t) = P(X_t = \mathcal{I}_0 \mid X_{t-1} = \mathcal{I}_1, \ldots, X_{t-R} = \mathcal{I}_R) \tag{2}$$

where each state $\mathcal{I}_k \in \mathcal{X}$ represents which ordered subset of size $B$ has been sampled at round $k$. For example, suppose clients $1$ to $B$ are sampled during the current round. $(1, 2, \ldots, B)$ and $(2, 1, 3, \ldots, B)$ are two different states, although the probability of these two states to appear is the same. The reason we consider this ordered case is that it allows us to cleanly define the probability of client $i$ to be sampled (which is the marginal distribution of $P(X_t)$) by noting that $P(i \text{ to be sampled at round } t) = \sum_{i_2, \ldots, i_B} P(X_t = (i, i_2, \ldots, i_B))$. Here we calculate the probability of client $i$ appearing as the first element in the ordered set $X_t$. The probability of $i$ being sampled in any position would need an additional scaling factor of $B$. Since the scaling factor $B$ is the same for all clients and only the relative frequency across clients contribute towards any bias effect, ignoring this factor of $B$ would not affect the debiasing calculation.

The above high-order Markov chain equation 2 has some nice properties as summarized below (see Appendix C for proofs). The insights are important for deriving Theorem 2, which is the reason we formally present them here.

**Proposition 1.** *The $R$-th order Markov chain equation 2 maintains the following properties:*

*(1). The ordered sequence $(\mathcal{I}_0, \mathcal{I}_1, \ldots, \mathcal{I}_R)$ is non-repeated, meaning $\mathcal{I}_l \cap \mathcal{I}_k = \emptyset, \forall l \neq k$.*

*(2). For any non-repeated $(\mathcal{I}_0, \ldots, \mathcal{I}_R)$,*

$$P(X_t = \mathcal{I}_0 \mid X_{t-1} = \mathcal{I}_1, \ldots, X_{t-R} = \mathcal{I}_R) = \frac{p_{\mathcal{I}_0}}{\sum_{\mathcal{J} \in S^c_{\mathcal{I}_{1:R}}} p_{\mathcal{J}}} =: p_{(\mathcal{I}_1, \ldots, \mathcal{I}_R) \to \mathcal{I}_0}. \tag{3}$$

*Otherwise $P(X_t = \mathcal{I}_0 \mid X_{t-1} = \mathcal{I}_1, \ldots, X_{t-R} = \mathcal{I}_R) = 0$. Since $\mathcal{I}_k$ is a set with $B$ unique elements, we define $p_{\mathcal{I}_k} := \prod_{e \in \mathcal{I}_k} p_e, \forall \mathcal{I}_k$. $S^c_{\mathcal{I}_{1:R}}$ is the collection containing all $B$-permutations of $[N] \setminus \bigcup_{k=1}^{R} \mathcal{I}_k$.*

*(3). For $t \geq R - 1$, define $Y_t = (X_t, \ldots, X_{t-R+1}) \in \mathbb{R}^R$. Then $\{Y_t\}_{t=R-1}^{\infty}$ is a conventional Markov chain with its cardinality of the state space being $d(M, R)$, where $d(M, R) = \prod_{k=0}^{R-1} \sigma(B(M - k), B)$. Moreover its transition probability is*

$$P(Y_t = (\mathcal{I}_0, \mathcal{J}_1, \ldots, \mathcal{J}_{R-1}) \mid Y_{t-1} = (\mathcal{I}_1, \ldots, \mathcal{I}_R)) = \begin{cases} p_{(\mathcal{I}_1, \ldots, \mathcal{I}_R) \to \mathcal{I}_0} & , \quad \mathcal{J}_k = \mathcal{I}_k, k \in [R-1] \\ 0 & , \quad \text{otherwise} \end{cases} \tag{4}$$

*for any non-repeated $(\mathcal{I}_0, \ldots, \mathcal{I}_R)$.*

*(4). Define vector $u_{(\mathcal{I}_1, \ldots, \mathcal{I}_R)} \in \mathbb{R}^{d(M,R)}$ with $(\mathcal{I}_0, \mathcal{I}_1, \ldots, \mathcal{I}_{R-1})$-th entry as $P(Y_t = (\mathcal{I}_0, \mathcal{I}_1, \ldots, \mathcal{I}_{R-1}) \mid Y_{t-1} = (\mathcal{I}_1, \ldots, \mathcal{I}_R))$. Then, $u_{(\mathcal{I}_1, \ldots, \mathcal{I}_R)} \in \mathbb{R}_+^{\sigma(B(M-R),B)} \subset \mathbb{R}^{d(M,R)}$ and $u_{(\mathcal{I}_1, \ldots, \mathcal{I}_R)}[(\mathcal{I}_0, \ldots, \mathcal{I}_{R-1})] = p_{\mathcal{I}_0}(\sum_{\mathcal{J} \in S^c_{\mathcal{I}_{1:R}}} p_{\mathcal{J}})^{-1} > 0, \forall \mathcal{I}_0 \in S^c_{\mathcal{I}_{1:R}}$.*

*(5). Denote $v_{(\mathcal{J}_0, \ldots, \mathcal{J}_{R-1})} \in \mathbb{R}^{d(M,R)}$ with $(\mathcal{J}_1, \mathcal{J}_2, \ldots, \mathcal{J}_R)$-th entry as $P(Y_t = (\mathcal{J}_0, \ldots, \mathcal{J}_{R-1}) \mid Y_{t-1} = (\mathcal{J}_1, \mathcal{J}_2, \ldots, \mathcal{J}_R))$ Then, $v_{(\mathcal{J}_1, \ldots, \mathcal{J}_R)} \in \mathbb{R}_+^{\sigma(B(M-R),B)}$ and $v_{(\mathcal{J}_0, \ldots, \mathcal{J}_{R-1})}[(\mathcal{J}_1, \ldots, \mathcal{J}_R)] = p_{\mathcal{J}_0}(\sum_{\mathcal{J} \in S^c_{\mathcal{J}_{1:R}}} p_{\mathcal{J}})^{-1} > 0$ for any $\mathcal{J}_R \in S^c_{\mathcal{J}_{0:R-1}}$.*

Properties (1),(2) essentially state that clients to be sampled in the current round cannot be those who have not waited for $R$ rounds, which establish the equivalence of our Markov-chain modeling equation 2 and the participation pattern in Section 2. Property (3) means that we can augment our state space by considering $R$-length history to formulate an equivalent Markov chain $\{Y_t\}_{t=R}^{\infty}$ with order 1. The last two properties explicitly shows what entries are for each row and column of the transition probability matrix of the new Markov chain $\{Y_t\}_{t=R}^{\infty}$. Also since there are only $\sigma(B(M - R), B) \ll d(M, R)$ non-zero entries in every row and column, the transition matrix is sparse.

A main benefit of this Markov-chain modeling is allowing us to look into the probability of each client to be sampled as $t$ goes on. Specifically, given any $R$, denote $P_R \in \mathbb{R}^{d(M,R) \times d(M,R)}$ as the

transition probability matrix of the Markov chain $\{Y_t\}_{t=R}^{\infty}$ where its entry is given by equation 4. Let $\phi_R(t) \in \mathbb{R}^{d(M,R)}$ be the state distribution at round $t$ of the Markov chain $\{Y_t\}_{t=R}^{\infty}$ and $\eta_R(t) \in \mathbb{R}^N$ be the distribution of clients to be sampled at round $t$. We have the following evolution of distributions with respect to $t$:

$$\eta_R(t) = Q_R^T \phi_R(t), \ \ \phi_R(t+1) = P_R^T \phi_R(t) \tag{5}$$

for any initial distribution $\eta_R(0)$ and corresponding $\phi_R(0)$ such that $\eta_R(0) = Q_R^T \phi_R(0)$, where $Q_R = Q_{R,1} Q_{R,2}$ and $Q_{R,1} \in \mathbb{R}^{d(M,R) \times \sigma(N,B)}$ is defined by

$$[Q_{R,1}]_{(\mathcal{I}_1,\ldots,\mathcal{I}_R),\mathcal{J}} = \begin{cases} p_{(\mathcal{I}_1,\ldots,\mathcal{I}_R) \to \mathcal{J}} & , \ \{\mathcal{J}, \mathcal{I}_1, \ldots, \mathcal{I}_R\} \text{ non-repeated} \\ 0 & , \ \ \text{otherwise.} \end{cases}$$

and $Q_{R,2} \in \mathbb{R}^{\sigma(N,B) \times N}$ is defined by

$$[Q_{R,2}]_{\mathcal{J},j} = \begin{cases} 1 & , \ \mathcal{J} = (j, *) \\ 0 & , \ \text{otherwise,} \end{cases}$$

where $\mathcal{J} = (j, *)$ denotes that the first entry of $\mathcal{I}$ is $j$. We are particularly interested in the distribution of $\eta_R(t)$ as $t \to \infty$ because it helps us characterize the asymptotic performance of existing FL algorithms. From classical Markov chain literature, we know that if a Markov chain is irreducible and aperiodic (see formal definitions in Appendix B), it has a stationary distribution which is unique and strictly positive. We denote $\zeta_R = \lim_{t \to \infty} \phi_R(t)$ as the stationary distribution of Markov chain $P_R$ and we have

$$\zeta_R^T = \zeta_R^T P_R, \ \ \pi_R^T = \zeta_R^T Q_R. \tag{6}$$

where $\pi_R \in \mathbb{R}^N$ is marginal stationary distribution of clients to be sampled, i.e., the $i$-th entry of $\pi_R$ is given by $\pi_R^i = \lim_{t \to \infty} \eta_R^i(t)$. On the other hand, if the Markov chain is irreducible and periodic, we let $\zeta_R$ be the Perron vector[2], which is also strictly positive. We now show our Markov chain is irreducibile and (a)periodic to justify the definitions of $\zeta_R$ and $\pi_R$ in Lemma 1. The proof is in Appendix C.

**Lemma 1.** *The Markov chain $\{Y_t\}_{t=R}^{\infty}$ with transition matrix $P_R$ defined by equation 4 is irreducible for all $M \geq 1$ and $0 \leq R \leq M - 1$. Further, when $R \leq M - 2$, it is also aperiodic.*

We provide an example to illustrate the intuition of our Markov-chain model above, considering the case of $N = 4, B = 1, R = 2$, i.e., every round one client is sampled, then it has to wait for two rounds. For instance, if client 1 and client 2 are consecutively selected in the first two rounds, in the third round only client 3 or 4 can be selected with probabilities of $p_3/(p_3 + p_4)$ or $p_4/(p_3 + p_4)$ respectively. Then, the state $(2, 1)$ can only transition to $(3, 2)$ or $(4, 2)$, where the second index is sampled before the first one as is in equation 2. Similarly, if we are currently at state $(1, 4)$, the previous state has to be $(4, 3)$ or $(4, 2)$. One can easily check that Proposition 1 holds. To see how $\pi$ is calculated, we take the first entry of $\pi_R$ as an example:

$$\pi_R^1 = \zeta^{(2,3)} p_{(2,3) \to 1} + \zeta^{(2,4)} p_{(2,4) \to 1} + \zeta^{(3,2)} p_{(3,2) \to 1} + \zeta^{(3,4)} p_{(3,4) \to 1} + \zeta^{(4,2)} p_{(4,2) \to 1} + \zeta^{(4,3)} p_{(4,3) \to 1}$$

by noting that the remaining $p_{(i,j) \to 1} = 0$, if $i$ or $j = 1$.

The vectors in equation 6 characterize the final distribution according to which clients will be sampled when the communication round $t$ becomes infinitely large. In other words, each client $i$ is sampled with probability $\pi_R^i$ given some fixed $R$. Although $\pi_{M-1}$ is the uniform distribution no matter what $p_i$'s are (by observing that all clients follow a cyclic participation), we note that $\pi_R$ for $R < M - 1$ does not necessarily follow the uniform distribution, because $\{p_1, \ldots, p_N\}$ are arbitrary. This will be problematic in the sense that existing federated learning algorithms may no longer guarantee convergence to the correct and optimal solution of equation 1 no matter how many rounds of training are implemented. We call this phenomenon *the asymptotic bias induced by $\pi_R$*. We will characterize both empirically and theoretically this phenomenon in the next section.

---

[2]We say $v$ is the Perron vector of the transition matrix $P$ if $v^T = v^T P$, i.e., $v$ is right eigenvector of $P$ corresponding to eigenvalue 1 and $v^T \mathbf{1} = 1$.

## 4 ASYMPTOTIC BIAS UNDER NON-UNIFORM CORRELATED PARTICIPATION

In this section, we utilize the Markov chain model in the previous section to analyze asymptotic bias of existing federated learning algorithms caused by $R < M - 2$ and arbitrary $p_i$'s. In particular, we consider FedAvg with local gradient descent updates, i.e., at each round, a set $S_t$ with $|S_t| = B$ clients are sampled and after being selected client $i$ updates its model as

$$x_{t,0}^i = x_t, \ x_{t,k+1}^i = x_{t,k}^i - \alpha \nabla f_i(x_{t,k}^i), \ k = 0, \ldots, K-1 \tag{7}$$

where $x_t$ denotes the server's model at round $t$ and $x_{t,k}^i$ is the local model maintained by client $i$ at $k$-th iteration. The server then updates $x_{t+1} = \frac{1}{B} \sum_{i \in S_t} x_{t,K}^i$. We next show in the following that FedAvg may not converge to the desired optimal solutions of equation 1. Instead there may exist some error neighborhood, i.e., the asymptotic bias, that is related to $\pi_R$, even as $t$ goes to infinity. Before we formally deliver the result, two standard assumptions are needed.

**Assumption 1.** *There exists $G > 0$ such that $\|\nabla f_i(x) - \nabla F(x)\|^2 \leq G^2, \forall x$ and $\forall i \in [N]$.*

**Assumption 2.** *Each $f_i$ is $L$-smooth, i.e., $\|\nabla f_i(x) - \nabla f_i(y)\| \leq L\|x - y\|, \forall x, y$ and $\forall i \in [N]$.*

Then we are ready to state the convergence of FedAvg under correlated client participation (see Appendix F for the proof).

**Theorem 1.** *Suppose Assumptions 1,2 hold and assume $\|\nabla F(x)\| \leq D, \forall x$ with some finite $D > 0$. Then for any $T > 2\tau_{mix} \log \tau_{mix}$ choosing $\alpha = \tilde{\mathcal{O}}(1/(\tau_{mix}K\sqrt{T}))$, FedAvg with local updates equation 7 generates the trajectory $\{x_t\}_{t=0}^{T-1}$ satisfying*

$$\mathbb{E}\|\nabla F(\tilde{x}_T)\|^2 \leq \tilde{\mathcal{O}}\left(\frac{\tau_{mix}}{\sqrt{T}}\right) + \mathcal{O}\left(\frac{1}{T}\right) + \mathcal{O}\left(\left\|\pi_R - \frac{1}{N}\mathbf{1}_N\right\|_1^2\right), \tag{8}$$

*for any $0 \leq R < M - 1$, where $\tilde{x}_T$ is drawn uniformly from $x_0, \ldots, x_{T-1}$, $\tilde{\mathcal{O}}(\cdot)$ hides logrithmic factors, and $\tau_{mix}$ denotes the mixing time[3]of Markov chain equation 5. Moreover, the bias term $\mathcal{O}\left(\left\|\pi_R - \frac{1}{N}\mathbf{1}_N\right\|_1^2\right)$ shown in equation 8 is unavoidable.*

Theorem 1 implies that without any debiasing technique, FedAvg can only converge to a solution with unavoidable asymptotic bias which is measured by the distance between $\pi_R$ (defined in equation 6) and the uniform distribution. Except for $R = M-1$, where $\pi_{M-1}$ is the uniform distribution, for $R \leq M - 2$, there is generally some gap between $\pi_R$ and $(1/N)\mathbf{1}_N$, which shows that FedAvg may fail to perform under correlated client participation. However, if $\pi_R$ is not too far away from the uniform distribution, we expect FedAvg to converge to a solution reasonably close to the optimal solution of equation 1. We next investigate what factors influence the distance from $\pi_R$ to the uniform distribution. We find that one factor is the spread among $p_i$'s. Stated by the following proposition, if all $p_i$'s are equal, no gap between $\pi_R$ and $(1/N)\mathbf{1}_N$ exists (see Appendix D for the proof).

**Proposition 2.** *Suppose $p_1 = p_2 = \cdots = p_N = \frac{1}{N}$. Then for any $0 \leq R \leq M - 1$, $\pi_R = \frac{1}{N}\mathbf{1}_N$.*

When $p_i$'s are not equal to each other, we turn to understand how $R$ affect $\pi_R$. In fact, we empirically observe that $\pi_R$ approaches the uniform distribution as $R$ increases. This key observation is illustrated in Figure 1. We consider the case where $N = 500, B = 1$ and assign each client a random $p_i > 0$. We then calculate $\pi_R$ for each $R$ ranging from 0 to $N - 1$ and measure its distance from the uniform distribution. As shown in the figure, increasing $R$ causes $\pi_R$ moving towards the uniform distribution. One explanation for this observation is that when $R$ becomes larger, fewer clients are ready to be sampled in the current round, because many clients have not waited for enough rounds and hence are not available. Rather than dictated by the availability probability $p_i$'s, which is the case for a small $R$ and many available clients, here the sampling process is mostly determined by the waiting requirement. In the extreme case, when $R = M - 1$, at each round, only $B$ clients are available, hence all clients are sampled with equal frequency. Another point suggested by this observation is that we can choose a large minimum separation $R$ in the practical scenario to reduce the asymptotic bias for existing FL algorithms, even with unknown $p_i$'s.

---

[3]Please refer to Appendix B for the formal definition of the mixing time.

---

**Algorithm 1** Debiasing FedAvg for correlated client participation

---

1: **Input:** initial point $x_0$, stepsizes $\{\alpha\}$, some $\tau > 0$, $\lambda_0 = \mathbf{0}_N$, $t_i = 0, \forall i \in [N]$ for each client
2: **for** $t = 0, 1, \ldots, T$ **do**
3:     A batch of clients $S_t$ with size $|S_t| = B$ is selected. The server sends current $t$ and model $x_t$ to clients in $S_t$.
4:     **for** $i \in S_t$ in parallel **do**
5:         Each client sets $t_i \leftarrow t_i + 1$ and calculates $\lambda_t^i = \frac{t_i}{(t+1)B}$ and $\nu_t^i = \frac{1}{\lambda_t^i N}$.
6:         **for** $k = 0, 1, \ldots, K - 1$ **do**
7:             Client $i$ updates its local model by

$$x_{t,k+1}^i = x_{t,k}^i - \alpha \nu_t^i \nabla f_i(x_{t,k}^i). \tag{9}$$

8:         **end for**
9:     **end for**
10:    The server updates its model $x_{t+1} = \frac{1}{B} \sum_{i \in S_t} x_{t,K}^i$.
11: **end for**
12: **Output:** $\tilde{x}_T$ sampled uniformly from $\{x_t\}_{t=0}^{T-1}$

---

The above empirical observation verifies the formal theorem that characterizes the debiasing effect of increasing minimum separation $R$ in Theorem 2. (see Appendix D for the proof).

**Theorem 2.** *Given a set of $p_i$'s, with at least one element $p_i \neq \frac{1}{N}$. Without loss of generality, let $p_1, \ldots, p_B$ be the $B$ smallest values among all $p_i$'s. Define $q_B := \sum_{j=1}^B p_j$, then $q_B < 1/M$. There exists a $\bar{\delta} > 0$, such that if any size-$B$ batch of clients $\mathcal{B}_j$ picking from $[N] \setminus [B]$, $\delta_j := |\sum_{l \in \mathcal{B}_j} p_l - \frac{1-q_B}{M-1}| \leq \bar{\delta}$, then $\pi_R$ converges to a neighborhood of $\frac{1}{N}\mathbf{1}_N$ characterized by $\{\pi \mid \|\pi - \frac{1}{N}\mathbf{1}_N\|_1 = \mathcal{O}(N^{-1})\}$ as $R$ ranging from 0 to $M - 1$. When $R = M - 1$, $\pi_{M-1}$ is the uniform distribution supported on $[N]$.*

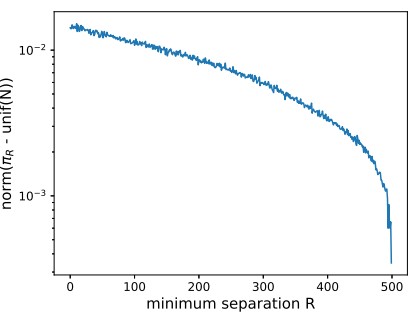

Figure 1: $\|\pi_R - \mathbf{1}_N/N\|_1$ as $R$ increases ($N = 500$, $B = 1$)

Theorem 2 states that when the availability probabilities $p_i$'s of clients are not too far away from each other or when $B$ is relatively large (i.e., $\delta_j$'s are small for all $j$), and when the total number of clients $N$ is large, $\pi_R$ approaches the uniform distribution as $R$ increases. It is worth noting that practically when the requirements in Theorem 2 are not strictly satisfied, the effect of increasing $R$ on $\pi_R$ can be still observed as shown in Figure 1.

## 5   Debiasing FedAvg and its convergence

As we discuss in the previous section, existing federated learning algorithms like FedAvg cannot guarantee convergence to the correct optimal solution if $R \leq M - 2$ and $p_i$'s are arbitrary. Although we can reduce the asymptotic bias caused by $\pi_R$ by increasing $R$, it may still be problematic under some particular circumstances. Clients have intermittent and non-uniform availability, and forcing a large minimum separation $R$ in practice may cause significant slowdown of the training in the FL system due to the small number of available clients. The minimum separation $R$ can be relatively small and the $p_i$'s can be very different from each other, which then suggests by Figure 1 and Theorem 2, $\pi_R$ can be far from the uniform distribution, making the asymptotic bias non-negligible. We next design a debiasing process that can be easily integrated into the existing federated learning algorithms to address asymptotic bias. Our proposed algorithm based on FedAvg is given by Algorithm 1.

The main difference between our algorithm and vanilla FedAvg lies in the stage of local updates (Lines 5 and 7). Specifically, we require each client to maintain an estimator of its corresponding

component of $\pi_R$, which is only updated when the client is sampled. This estimator is later used to scale the gradient step during the local update. The estimator is designed by counting the times the client has been sampled and then used to compute the running empirical frequency of the client's participation. Recall that $\pi_R^i$ represents the frequency of client $i$ to be selected when $t$ is large enough (i.e., when the Markov chain equation 5 becomes steady, meaning $\phi_R(\infty) = \zeta_R$). If we reweigh the local objective function $f_i$ by $\frac{1}{\pi_R^i}$ (corresponding to $\nu_t^i = \frac{1}{\pi_R^i N}$ in equation 9), this weighting cancels the asymptotic bias introduced by unbalanced sampling, which drives the trajectory of the server's models towards the correct solution of equation 1. If we know $\pi_R^i$ for every client in prior, the above-mentioned reweighting method provides us with unbiased solutions. Then, $\lambda_t^i$ serves as a role to iteratively approximate $\pi_R^i$ round by round, which yields Algorithm 1. Also note that Algorithm 1 reduces to FedAvg if fixing $\lambda_t^i = 1/N, \forall i \in [N]$. This shows the advantage of our algorithm: it is computationally cheap in the sense that each client only maintains two additional scalars ($\lambda_t^i$ and $\nu_t^i$) and can be easily embedded with existing algorithms by just multiplying the learning rates by $\nu_t^i$. We note that other federated algorithms suffering from asymptotic bias due to non-uniform sampling could also benefit from our debiasing technique based on simple counting.

However, formally characterizing the convergence of $\nu_t^i$ to $\frac{1}{\pi_R^i N}$ remains challenging due to the samples of clients are not independent across different rounds. In particular, the clients sampled in the current round may affect those in the future, which makes the conventional concentration tools and law of large numbers not applicable. To address this challenge, we carefully analyze the transition of the Markov chain equation 5 and its influences on the marginal distribution of clients to be sampled to conclude that $\lambda_t^i$ is an unbiased estimate of $\pi_R^i$ asymptotically. Then, we further leverage the fact that the Markov chain is irreducible as stated in Lemma 1 to show that $\lambda_t^i$ is almost surely strictly positive even $t$ is infinite, concluding the convergence of $\nu_t^i$ to $\frac{1}{\pi_R^i N}$, as summarized in Lemma 2 (see Corollary 2 in Appendix G for the proof).

**Lemma 2.** *Given $\lambda_0 = \mathbf{0}_N$, then $\nu_t^i, \forall i \in [N]$ in Algorithm 1 satisfies*

$$\mathbb{E}\|\tilde{\nu}_t\|_\infty^2 \leq \mathcal{O}\left(\frac{\tau_{mix}}{t}\right)$$

*for any $t > 0$, where $\tilde{\nu}_t^i = \nu_t^i - \frac{1}{\pi_i N}$ and $\tilde{\nu}_t = (\tilde{\nu}_t^1, \ldots, \tilde{\nu}_t^N)$; $\tau_{mix}$ is the mixing time of Markov chain equation 5.*

Based on the above, we can achieve the following convergence result of Algorithm 1 (see Appendix G for the proof).

**Theorem 3.** *Suppose Assumptions 1 and 2 hold. For any $0 \leq R < M-1$ and $T > c^\dagger \tau_{mix} \log \tau_{mix}$ (with $c^\dagger$ being some constant), choosing $\alpha = \tilde{\mathcal{O}}(1/(\tau_{mix} K \sqrt{T}))$, the output of Algorithm 1 satisfies*

$$\mathbb{E}\|\nabla F(\tilde{x}_T)\|^2 = \tilde{\mathcal{O}}\left(\frac{\tau_{mix}}{\sqrt{T}}\right) + \mathcal{O}\left(\frac{1}{T}\right)$$

*where $\tilde{x}_T$ is defined as that in Theorem 1; $\tau_{mix}$ is the mixing time of Markov chain equation 5.*

Comparing to Theorem 1, no bounded gradient assumption is needed to reach the convergence of our algorithm. Unlike the result in (Cho et al., 2023) where clients are forced to participate in the system cyclically, our bound shown in Theorem 3 does not grow as the number of clients increases. Particularly, for the bounds in (Cho et al., 2023) to be non-vacuous, the total number of communication round $T$ should be proportional to the number of clients, which could be hard to satisfy in practice especially when client number is super large. To prove Theorem 3 we critically rely on the fact that the Markov chain equation 5 is aperiodic to make analysis go through. That is to say our bound does not suit for $R = M - 1$, which is the limitation of our analysis. However, since $R = M - 1$ is the cyclic case, where the Markov chain follows much nicer structure, one may be able to get a better bound (Cho et al., 2023).

We remark that our convergence result achieves nearly the same order of rate as Markov-sampling SGD literature (Beznosikov et al., 2024; Even, 2023) (where rates of $\mathcal{O}(\sqrt{\tau_{mix}}/\sqrt{T} + \tau_{mix}/T)$ are obtained). However, their analysis only suits for the first-order Markov chain and no debiasing results are presented, while our results generalize to high-order Markov chain and guarantee approaching unbiased solutions. We note that utilizing variance-reduced techiques may accelerate the

convergence rate for Markov-sampling SGD (Even, 2023). Then whether variance reduction can be used in our problem to design faster algorithms would be an interesting future direction.

It is worth noting that although a uniform minimum separation $R$ for all clients is placed throughout the paper, we also allow each client maintains its own specific $R_i, \forall i \in [N]$. In this more general case, we could still use the same modeling technique as in Section 3 where the order of the Markov chain is chosen to be an upper bound of all $R_i$'s (e.g. $\max_i R_i$). Then Theorems 1 and 3 can be obtained without any modification as the analysis stays valid for any irreducible and aperiodic Markov chain. However, Theorem 2 becomes tricky in this case as our proof highly relies on nice properities of the Markov chain summarized by Proposition 1 which now cease to hold. Therefore, more advanced mathematical tools might be needed in order to obtain similar statements as Theorem 2 when clients have various $R_i$'s.

# 6 NUMERICAL RESULTS

In this section, we provide numerical experiments to illustrate our theoretical results. In particular, we compare vanilla FedAvg with our proposed algorithm (Algorithm 1) under non-uniform and correlated client participation described in Section 2. For simplicity, we partition the $N$ clients into $M$ groups and exactly one group of clients are selected at each round to fully participate in the system. Here we choose $N = 100, M = 20$. Since all clients in the same group participate in the system together once being sampled, we only need to associate availability probabilities to each group, where $p_i \propto i^{-1.5}, i \in [M]$ is a long-tailed distribution.

**Synthetic dataset.** We test Vanilla FedAvg and Debiasing FedAvg (Algorithm 1) under a synthetic dataset constructed following (Sun & Wei, 2022): for each client $i$, $A_i \in \mathbb{R}^{n_i \times d}$ is the feature matrix, where $n_i$ is the number of local samples and $d$ is the feature dimension. Every entry of $A_i$ is generated by a Gaussian distribution $\mathcal{N}(0, (0.5i)^{-2})$. We then generate $b_i \in \mathbb{R}^{n_i}$, the labels of client $i$, by first generating a reference point $\theta_i \in \mathbb{R}^d$, where $\theta_i \sim \mathcal{N}(\mu_i, I_d)$. And $\mu_i$ is drawn from $\mathcal{N}(\alpha, 1)$ with $\alpha \sim \mathcal{N}(0, 100)$. Then $b_i = A_i \theta_i + \epsilon_i$ with $\epsilon_i \sim \mathcal{N}(0, 0.25 I_{n_i})$. We set $d = 20, n_i = 100, \forall i \in [N]$. And we define $f_i(x) = \frac{1}{n_i} \sum_{j=1}^{n_i} \log(\frac{1}{2}(\langle A_i[j, :], x \rangle + b_i[j])^2 + 1)$ where $A_i[j, :]$ represents the $j$-th row of $A_i$ and $b_i[j]$ is the $j$-th entry of $b_i$. The outcomes are shown in Figures 2a,2b, where Figure 2a shows that Vanilla FedAvg suffers from bias which can be mitigated by increasing $R$, and Figure 2b shows that Debiasing FedAvg effectively reduces bias no matter what value of $R$ is set.

**MNIST dataset.** We also test our proposed algorithm under the MNIST dataset. Each client maintains a three-layer fully-connected neural network for training. All learning rates are chosen to be with the order of $\mathcal{O}(10^{-3})$. In Figure 3c, we compare Debiasing FedAvg with Vanilla FedAvg and FedVARP (Jhunjhunwala et al., 2022), and Debiasing FedAvg can effectively mitigate the bias effect. Another interesting empirical observation is that increasing $R$ can possible fasten the speed of both Debiasing and Vanilla FedAvg (as shown by Figures 3a,3b). This is yet not characterized by our theoretical demonstration. Here we conjecture that larger $R$ corresponds to smaller mixing time $\tau_{mix}$ and hence faster rate as the bounds in Theorems 1,3 scale with respect to $\tau_{mix}$. We provide more detailed and intuitive discussions in Appendix I.

# 7 LIMITATIONS

Our Markov-chain framework works for that all clients share the same static minimum separation $R$ and in the last paragraph of Section 5 we further allow static client-specific $R_i$'s. However, more practical scenarios call for even time-varying $R$, which lies outside the scope this paper. In Theorem 2, we force the availability probabilities $p_i$'s not too far away from each other to make the theory hold, while this assumption is not required in practice. Moreover, Theorems 1,3 do not enjoy speedup in the number of clients as FL literature when clients are uniformly sampled. We believe it is mainly due to the non-uniformity and time-correlation of the client sampling process and more advanced mathematical tools are needed to show speedup in the Markov setting.

Many of the challenges in cross-device federated learning can also be addressed by system design in addition to algorithm design. For example, incorporating trusted execution environment (TEEs) in

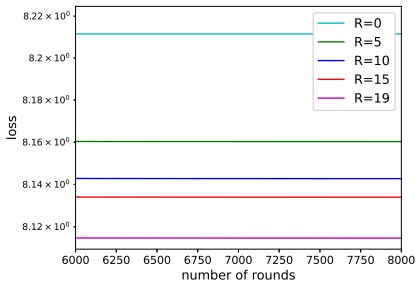

(a) FedAvg under different $R$ after convergence (synthetic data)

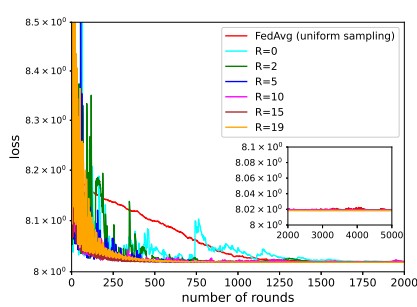

(b) Debiasing FedAvg under different $R$ (synthetic data)

Figure 2: Experiments on synthetic dataset. (a) The training loss of Vanilla FedAvg (after convergence) with different $R$ is shown. Larger $R$ leads to smaller bias. (b) Debiasing FedAvg is tested under different values of $R$, where the red line represents Vanilla FedAvg when clients are sampled under an oracle uniform distribution. The subfigure on the right shows that all curves reach unbiased objective after convergence, indicating that the asymptotic bias is effectively canceled.

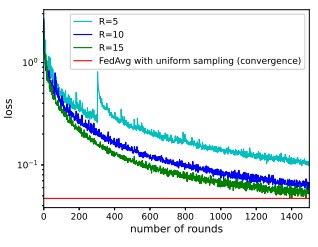

(a) Debiasing FedAvg under different $R$ (MNIST)

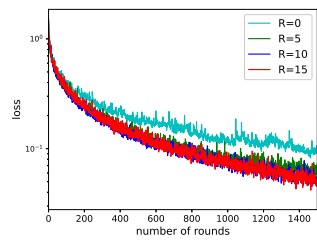

(b) FedAvg under different $R$ (MNIST)

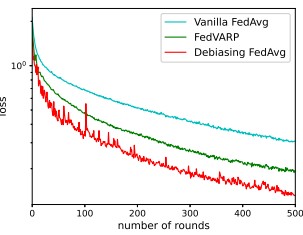

(c) FedAvg, FedVARP, Debiasing FedAvg when $R = 8$ (MNIST)

Figure 3: Experiments on MNIST. (a) The convergence of our Debiasing FedAvg under different client minimum separation $R$ configurations. The red horizontal line is the convergence value of the objective function by vanilla FedAvg when clients are sampled under an oracle uniform distribution. Our Debiasing FedAvg converges to the unbiased objective with larger $R$ converges faster. (b) For Vanilla FedAvg, increasing $R$ causes smaller bias. (c) When $R = 8$, Vanilla FedAvg, FedVARP and Debiasing FedAvg are compared. Note that both Vanilla FedAvg and FedVARP are designed only for uniform client sampling and hence are significantly affected by bias from client participation.

the FL system (Huba et al., 2022; Daly et al., 2024) can potentially provide more control on client sampling.

## 8 CONCLUSION

In this paper, we consider FL with non-uniform and correlated client participation, where every client must wait as least $R$ rounds (minimum separation) before participating again, and each client has their own availability probability. A high-order Markov chain is introduced to model this practical scenario. Based on this Markov-chain modeling, we are able to study the convergence performances of existing FL algorithms. Due to the effect of non-uniformity and time correlation, FL algorithms can only converge with asymptotic bias, which can be reduced by increasing minimum separation $R$ as shown by our empirical and theoretical results. Finally, we propose a debiasing algorithm for FedAvg that guarantee convergence to unbiased solutions given arbitrary non-uniformity and minimum separation $R$.

ACKNOWLEDGMENTS

This work was supported in part by NSF grants CCF-2045694, ECCS-2216970, CMMI-2024774, CNS-2112471, CPS-2111751, ONR N00014-23-1-2149, and a Google Research Scholar Award to GJ.

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

## A   RELATED WORK

**Non-uniform & correlated client participation.**    There is a recent surge of efforts to investigate FL with non-uniform client participation both from theoretical and empirical perspectives. Earlier work presumes that clients are sampled by the server uniformly, which guarantees the global model held by the server is an unbiased estimate as that in the full participation setting and hence allows extension of convergence results for the full-participation setting to the partial-participation setting (Jhunjhunwala et al., 2022; Karimireddy et al., 2019; Yang et al., 2021; Bian et al., 2024).  The above-mentioned uniform participation is, however, far from the reality as clients may have their intrinsic sampling probabilities $p_i$'s that are non-uniform due to, for example, intermittent availability resulting from practical constraints. Recent works analyzed the convergence behaviors of FL algorithms when such $p_i$'s are known as a prior or controllable (Wang & Ji, 2022; Karimireddy et al., 2019; Chen et al., 2020; Fraboni et al., 2021). However, pointed out by (Bonawitz et al., 2019; Wang et al., 2021), client participation pattern can highly depend on the underlying system characteristics, which is thus hard to know or control. As characterized by (Wang & Ji, 2022; Xiang et al., 2024), such unknown and non-uniform participation statistics causes a bias in the model updates as more frequently participating clients dominate the average update. In order to mitigate the effect of bias, (Patel et al., 2022; Ribero et al., 2022; Wang & Ji, 2023) introduced reweighting mechanisms combined with dynamically estimating client participation distributions. Such idea is also introduced in asynchronous distributed learning literature (Ram et al., 2009). Most works aiming at analyzing non-uniform participation, however, rely on the unrealistic assumption that every client participates in the system independently, which fails to capture practical scenarios where each client's participation is influenced by others across rounds(Kairouz et al., 2019; Eichner et al., 2019; Zhu et al., 2021). One interesting time-correlated participation pattern is that clients have to wait for at least $R$ (called minimum separation) rounds between consecutive participation (McMahan & Thakurta, 2022; Xu et al., 2023). In particular, imposing a minimum separation constraint has been empirically shown to benefit privacy preservation in FL applications (Kairouz et al., 2021; Choquette-Choo et al., 2023; Xu et al., 2023; Xu & Zhang, 2024). Instead, such time-correlated participation has not been fully investigated theoretically. The only work that partially captures the above case is (Cho et al., 2023) where the clients are forced to follow a cyclic participation, which is an extreme case of very large $R$. Therefore, in this paper we study convergence performances of FL algorithms under non-uniform and correlated client participation, which provides theoretical explanations for their empirical counterparts in practice.

**Stochastic optimization with Markov-sampling.**    Another line of related works is stochastic gradient-based optimization under Markov-sampling. Unlike classical stochastic optimization literature where i.i.d. samples are drawn during the training process (Allen-Zhu & Hazan, 2016; Allen-Zhu, 2017; Johnson & Zhang, 2013; Defazio et al., 2014), many contexts, including TD-learning and reinforcement learning (RL), require to optimize the objective function by utilizing samples generated by a Markov chain (Tsitsiklis & Van Roy, 1996; 1999; Bhatnagar et al., 2007; Sutton et al., 1999). Recently, the work (Even, 2023) provided convergence guarantees for SGD under Markov-sampling when the objectives are convex, strongly convex and non-convex. Then (Beznosikov et al., 2024) further proposed an accelerated method and generalized the analysis to variational inequalities. Both of them restrict on the first-order Markov chains. It has been shown by literature that gradient-based methods converge to the optimal solution of the objective induced by the stationary distribution of the underlying Markov chain (Even, 2023; Beznosikov et al., 2024). This indicates that the final solution is biased if the stationary distribution is non-uniform and existing literature cannot deal with such bias problem. In contrast, in this paper our analysis suits for higher-order Markov chains and the proposed algorithm enables the convergence to an unbiased solution without any information and constraint on the Markov chain and stationary distribution.

## B   PRELIMINARIES OF MARKOV CHAINS

In this section, we summarize several notions and properties of the conventional Markov chain (i.e., first-order Markov chain). We only focus on finite Markov chains, meaning the state space is finite. Note that for a finite Markov chain, we can use its transition matrix to uniquely represent it.

**Definition 2.** *Given a finite Markov chain with transition matrix $P$, we say it is irreducible if its induced graph is strongly connected, i.e., every state can be reached from every other state.*

Note that $[P^k]_{i,j}$ is the probability transiting from state $i$ to state $j$ with exactly $k$ steps, based on which we introduce the definition of aperiodic and periodic Markov chains.

**Definition 3.** *The period of state $i$ is the greatest common divisor (g.c.d.) of the set $\{k \in \mathbb{N} \mid [P^k]_{i,i} > 0\}$. If every state has period $1$ then the Markov chain is aperiodic, otherwise it is periodic.*

In order words, the period of state $i$ can be achieved by calculating the g.c.d. of the number of steps starting from $i$ and returning back. If the Markov chain is also irreducible, we have the following.

**Lemma 3.** *If the Markov chain is irreducible, every state has the same period.*

Next important result states the convergence of the Markov chain.

**Lemma 4.** *Suppose a finite Markov chain with transition matrix $P$ is irreducible and aperiodic. Then, there exist some $\rho \in (0, 1)$ and $C > 0$ such that*

$$\max_x \|P^k(x, \cdot) - \pi\|_{TV} \le C\rho^k$$

*where $\pi$ is the unique, strictly positive stationary distribution; $\|\cdot\|_{TV}$ denotes the total variation.*

Lemma 4 implies that starting from any initial distribution, the Markov chain converges to the stationary distribution at linear rate. Without confusion, we denote $d_{TV}(P^k, \mathbf{1}\pi^T) = \max_x \|P^k(x, \cdot) - \pi\|_{TV}$. Note that $d_{TV}(P^k, \mathbf{1}\pi^T) = \frac{1}{2}\|P^k - \mathbf{1}\pi^T\|_\infty$. Then, we define the mixing time of the chain.

**Definition 4.** *Given any $\epsilon > 0$, the mixing time $t_{mix}(\epsilon)$ is defined as $t_{mix}(\epsilon) := \inf\{l \ge 1 \mid d_{TV}(P^l, \mathbf{1}\pi^T) \le \epsilon\}$. Conventionally, we denote $\tau_{mix} = t_{mix}(1/4)$.*

**Lemma 5.** *We have the following statements:*

*(1). $d_{TV}(P^{t+1}, \mathbf{1}\pi^T) \le d_{TV}(P^t, \mathbf{1}\pi^T), \forall t \ge 0$.*

*(2). For $k \ge 2$, $t_{mix}(2^{-k}) \le (k-1)\tau_{mix}$.*

*(3). Moreover,*

$$\sum_{k=0}^{T} d_{TV}(P^k, \mathbf{1}\pi^T) \le c_0 \tau_{mix}, \ \ \forall T \ge 0$$

*for some constant $c_0 > 0$.*

*Proof.* The first two claims are shown in (Levin & Peres, 2017). To see the third claim, we note that

$$\sum_{k=0}^{T} d_{TV}(P^k, \mathbf{1}\pi^T) \le \sum_{k=0}^{\infty} d_{TV}(P^k, \mathbf{1}\pi^T)$$

$$\le \sum_{l=0}^{\tau_{mix}} d_{TV}(P^l, \mathbf{1}\pi^T) + \sum_{k=2}^{\infty} \sum_{l=t_{mix}(2^{-k})+1}^{t_{mix}(2^{-(k+1)})} d_{TV}(P^l, \mathbf{1}\pi^T)$$

$$\le d_{TV}(P, \mathbf{1}\pi^T)\tau_{mix} + \sum_{k=2}^{\infty} (t_{mix}(2^{-(k+1)}) - t_{mix}(2^{-k}))2^{-k}$$

$$\le d_{TV}(P, \mathbf{1}\pi^T)\tau_{mix} + \sum_{k=2}^{\infty} k2^{-k}\tau_{mix}$$

$$\le d_{TV}(P, \mathbf{1}\pi^T)\tau_{mix} + 2\tau_{mix}$$

which completes the proof with $c_0 = d_{TV}(P, \mathbf{1}\pi^T) + 2$. $\square$

## C  PROOFS OF PROPOSITION 1 AND LEMMA 1

### C.1  PROOF OF PROPOSITION 1

Property (1) follows from equation 2, where by definition within $R + 1$ rounds, the clients cannot participate in the system twice or more. Property (2) follows from the fact that every client is

sampled with probability proportional to its availability probability $p_i$ if it has waited for $R$ rounds. Property (3) then directly follows from the definition of the first-order Markov chain and Properties (1), (2). Properties (4) and (5) are due to the observation that every row and column has exactly $\sigma(B(M-R), B)$ non-zero entries due to Property (3).

### C.2 PROOF OF LEMMA 1

It is obvious that the Markov chain is irreducible in the sense that all ordered sequences $(\mathcal{I}_1, \ldots, \mathcal{I}_R)$ can be observed due to every client has strictly positive probability to be selected. To see that it is aperiodic for $R \leq M - 2$, we only need to show that starting from the state $(\mathcal{I}_1, \ldots, \mathcal{I}_R)$ where $\mathcal{I}_k = ((k-1)B+1, \ldots, kB), k = 1, \ldots, R$, both $R+1$ steps and $R+2$ steps can be possibly taken such that the first return happens, which implies aperiodicity. This is because if a Markov chain is irreducible, all the states have the same period by Lemma 3.

Then, we consider the following two constructed sequence. Let $h_1 = (\mathcal{I}_1, \ldots, \mathcal{I}_R, \mathcal{I}_{R+1}, \mathcal{I}_1, \ldots, \mathcal{I}_R)$ for state $\mathcal{I}_{R+1} = (RB + 1, \ldots, (R + 1)B)$, where the length of $h_1$ is $2R + 1$. Denote $h_1[k]$ as the entry at the $k$-th position. We construct the sequence $\{Y_t, Y_{t+1}, \ldots, Y_{t+R}\}$ as $Y_{t+k-1} = (h_1[k \bmod (2R+1)], \ldots, h_1[(k + R - 1) \bmod (2R + 1)]), k = 1, \ldots, 2R+1$, i.e., starting from $(\mathcal{I}_1, \ldots, \mathcal{I}_R)$ exactly $R+1$ steps are taken to firstly return. Similar to the definition of $h_1$, let $h_2 = (\mathcal{I}_1, \ldots, \mathcal{I}_R, \mathcal{I}_{R+1}, \mathcal{I}_{R+2}, \mathcal{I}_1, \ldots, \mathcal{I}_R)$ with its length $2R + 2$ and state $\mathcal{I}_{R+2} = ((R + 1)B + 1, (R + 2)B)$. We then construct the sequence $\{Y_t, \ldots, Y_{t+R+1}\}$ as $Y_{t+k-1} = (h_2[k \bmod (2R+2)], \ldots, h_2[(k + R - 1) \bmod (2R + 2)]), k = 1, \ldots, 2R+2$, which then suggests exactly $R + 2$ steps are required to return back to $(\mathcal{I}_1, \ldots, \mathcal{I}_R)$. Combining these two cases leads to the Markov chain is aperiodic for any $R \leq M - 2$.

## D PROOFS OF PROPOSITION 2 AND THEOREM 2

### D.1 PROOF OF THEOREM 2

Let us first consider the case when $B = 1$ and given $p_1 > 0$, $p_i = \frac{1-p_1}{N-1}, \forall i = 2, \ldots, N$. Then, for any $0 < R \leq N - 1$ and any $(j_0, \ldots, j_{R-1})$, pick an arbitrary $j_R \in \{j_0, \ldots, j_{R-1}\}^c$. By denoting $b_R = b(P_R[\cdot, (j_0, \ldots, j_{R-1})])$, $b_{R+1} = b(P_{R+1}[\cdot, (j_0, \ldots, j_R)])$ (which are the column sums for each column of $P_R$ and $P_{R+1}$, respectively) and letting $S_R := \{j_0, \ldots, j_{R-1}\}$, $S_{R+1} := \{j_0, \ldots, j_R\}$ for notation simplicity. By observing that when $\pi_R$ is exactly the uniform distribution, the sum of $P_R$ for each column is exactly one, we then tend to prove that the column sum of $P_R$ asymptotically approaches one as $R$ increases. We have four cases.

Case I: $j_0 = \{1\}$. Then, for any $0 \leq R \leq N - 2$, utilizing last two properties in Proposition 1,

$$
b_{R+1} - b_R = p_1 \sum_{k \in S_{R+1}^c} \left(p_1 + \sum_{i \in S_{R+1}^c} p_i - p_k\right)^{-1} - p_1 \sum_{k \in S_R^c} \left(p_1 + \sum_{i \in S_R^c} p_i - p_k\right)^{-1}
$$

$$
= p_1 \sum_{k \in S_{R+1}^c} \left(p_1 + \frac{1 - p_1}{N - 1}(N - R - 2)\right)^{-1} - p_1 \sum_{k \in S_R^c} \left(p_1 + \frac{1 - p_1}{N - 1}(N - R - 1)\right)^{-1}
$$

$$
= p_1(N - R - 1)\left(p_1 + \frac{1 - p_1}{N - 1}(N - R - 2)\right)^{-1} - p_1(N - R)\left(p_1 + \frac{1 - p_1}{N - 1}(N - R - 1)\right)^{-1}
$$

Let $r = N - R - 1$. We simply $b_R$ as

$$
b_R = \frac{p_1 r}{p_1 + \frac{1-p_1}{N-1}(r-1)} = \frac{p_1(N-1)}{1 - p_1} + \frac{p_1\left(1 - \frac{p_1(N-1)}{1-p_1}\right)}{p_1 + \frac{1-p_1}{N-1}(r-1)}
$$

Then,

$$
b_{R+1} - b_R = p_1\left(1 - \frac{p_1(N-1)}{1-p_1}\right)\left(\frac{1}{p_1 + \frac{1-p_1}{N-1}(r-1)} - \frac{1}{p_1 + \frac{1-p_1}{N-1}r}\right)
$$

$$
= \frac{p_1(1-p_1)}{N-1}\left(1 - \frac{p_1(N-1)}{1-p_1}\right)\left(p_1 + \frac{1-p_1}{N-1}(r-1)\right)^{-1}\left(p_1 + \frac{1-p_1}{N-1}r\right)^{-1}
$$

which is strictly positive for $p_1 < 1/N$ for all $0 \le R \le N - 2$.

Case II: $\{1\} \in S_{R+1}^c$. Then, we obtain $p_{j_0} = \frac{1-p_1}{N-1}$ and hence

$$
\begin{aligned}
b_R/p_{j_0} &= (p_{j_0} + \frac{1-p_1}{N-1}(N-R-1))^{-1} + (N-R-1)(p_1 + \frac{1-p_1}{N-1}(N-R-1))^{-1} \\
&= \frac{N-1}{(1-p_1)(r+1)} + r(p_1 + \frac{1-p_1}{N-1}r)^{-1} \\
&= \frac{N-1}{(1-p_1)(r+1)} + \frac{N-1}{1-p_1} \frac{p_1(N-1)}{1-p_1} \frac{1}{p_1 + \frac{1-p_1}{N-1}r}
\end{aligned}
$$

where we let $r = N - R - 1$. Then, denoting $\bar{p} = \frac{1-p_1}{N-1}$ and $\alpha = p_1/\bar{p}$ yields

$$
\begin{aligned}
(b_{R+1} - b_R)/p_{j_0} &= \frac{1}{\bar{p}} \frac{1}{r(r+1)} - \frac{p_1}{\bar{p}^2(r+\alpha-1)(r+\alpha)} \\
&= \frac{(r+\alpha)(r+\alpha-1) - \alpha r(r+1)}{\bar{p}r(r+1)(r+\alpha)(r+\alpha-1)} \\
&= \frac{(1-\alpha)r^2 + (\alpha-1)r + \alpha(\alpha-1)}{\bar{p}r(r+1)(r+\alpha)(r+\alpha-1)} \\
&= \frac{(1-\alpha)(r^2 - r - \alpha)}{\bar{p}r(r+1)(r+\alpha)(r+\alpha-1)}.
\end{aligned}
$$

Note that when $p_1 < 1/N$, $\alpha < 1$, which indicates $b_{R+1} - b_R > 0, \forall 0 \le R \le N - 3$ by observing $r^2 - r - \alpha \ge 0$. Moreover, note that $b_R > 1, \forall R \le N - 2$ in this case by

$$
b_R = \frac{1}{r+1} + 1 - \frac{\alpha}{r+\alpha} = \frac{(1-\alpha)r}{(r+1)(r+\alpha)} + 1 > 1
$$

for $\alpha < 1$. And a straightforward calculation gives $b_{N-2} < \frac{3}{2}$, which then indicates $|b_R - 1| < \frac{1}{2}, \forall R \le N - 1$.

Case III: $\{1\} \in S_R^c$ and $\{1\} \notin S_{R+1}^c$. In this case, $p_{j_0} = \frac{1-p_1}{N-1} = \bar{p}$. Then, a simple calculation gives

$$
(b_{R+1} - b_R)/p_{j_0} = \frac{1}{\bar{p}} \frac{(\alpha-1)r}{(r+1)(r+\alpha)} < 0
$$

when $p_1 < 1/N$.

Case IV: $\{1\} \notin S_R^c$. Then, all the clients are available in both $S_R^c$ and $S_{R+1}^c$ have availability probability $\bar{p}$. Then, it is obvious that $b_R = 1, \forall 0 \le R \le N - 1$.

For Cases I, III and IV, we conclude that when $p_1 < 1/N$ and $p_i = \frac{1-p_1}{N-1}, i = 2, \dots, N, |b_{R+1} - 1| < |b_R - 1|, \forall 0 \le R \le N - 2$ by further noting that $b_{N-1} = 1$. By Case II, we then have all $|b_R - 1|$ converges to $[0, 0.5]$ as $R$ increases. Observe $b_{N-1} = 1$ corresponds to the case that $\zeta_{N-1}$ is exactly the uniform distribution and so is $\pi_{N-1}$. This indicates that $\pi_R$ converges to some neighborhood of the uniform distribution $\frac{1}{N}\mathbf{1}_N$. In order to characterize this neighborhood, we turn to carefully analyze Case II, i.e., $|b_R - 1| < 0.5$. Noting that Case II corresponds to at most $1 - R/N$ portion of columns in $P_R$ and so does $\pi_R$, therefore the neighborhood is characterized by $\{\pi \mid \|\pi - \frac{1}{N}\mathbf{1}_N\|_1 = \mathcal{O}(1/N)\}$.

Next, in order to prove the statement, we perturb each $p_i = \frac{1-p_1}{N-1}, i = 2, \dots, N$ by some scalar $\epsilon_i$ such that $\sum_{i=2}^N \epsilon_i = 0$. Note that $b_{R+1} - b_R$ is continuous in $(\epsilon_2, \dots, \epsilon_N)$ and so is $\pi_R$, which then implies that there exists some positive $\Delta > 0$ such that $b_{R+1} - b_R$ preserves the original properties as before the perturbation is added for all $|\epsilon_i| \le \Delta$. Therefore, we achieve the statement that $\pi_R$ converges to the neighborhood $\{\pi \mid \|\pi - \frac{1}{N}\mathbf{1}_N\|_1 = \mathcal{O}(1/N)\}$ when $B = 1$. Obtaining the statement for $B > 1$ follows the same technique by noting that we can always calculate the equivalent $\tilde{p}_i$ for each batch with size $B$. Specifically, given a batch of clients, say $\mathcal{B}_i$, then $\tilde{p}_i = \prod_{j \in \mathcal{B}_i} p_j / C$ with suitable normalization constant $C$ and we can then obtain the convergence of $\pi_R$ to a neighborhood of the uniform distribution by similar development.

## D.2 PROOF OF PROPOSITION 2

The proof of Proposition 2 is straightforward by observing that $b_R = 1, \forall R$ when $p_i = 1/N, \forall i \in [N]$. Then $\mathbf{1}^T P_R = \mathbf{1}^T, \forall R$ which indicates $\pi_R$ is always the uniform distribution.

## E INTERMEDIATE LEMMAS

In this section, we present some useful intermediate results under the following generalized setting: we consider a general global objective function defined as $F_w(x) := \sum_{i=1}^N w_i f_i(x)$ where $\sum_{i=1}^N w_i = 1$ and $w_i \geq 0, \forall i \in [N]$. And we consider the following local update

$$x_{t,k+1}^i = x_{t,k}^i - \alpha q_t^i \nabla f_i(x_{t,k}^i) \tag{10}$$

where $q_t^i = \frac{w_i}{y_t^i}$ for some positive sequence $y_t^i$. Note that the above update equation 10 is a generalized version of equation 9 in Algorithm 1. Then we have the following useful lemmas when forcing the update equation 10.

**Lemma 6.** *Under Assumption 1, we have for any $x$*

$$\|\nabla F_w(x) - \nabla F(x)\| \leq G$$
$$\|\nabla f_i(x) - \nabla F_w(x)\| \leq 2G, \ \forall i \in [N].$$

*Proof.* Note that Assumption 1 implies

$$\|\nabla F_w(x) - \nabla F(x)\| = \|\sum_{i=1}^N w_i(\nabla f_i(x) - \nabla F(x))\|$$

$$\leq \sum_{i=1}^N w_i \|\nabla f_i(x) - \nabla F(x)\|$$

$$\leq G \sum_{i=1}^N w_i = G.$$

Then, for any $i \in [N]$

$$\|\nabla f_i(x) - \nabla F_w(x)\| \leq \|\nabla f_i(x) - \nabla F(x)\| + \|\nabla F_w(x) - \nabla F(x)\| \leq 2G.$$

$\square$

**Lemma 7.** *Given any $t$ and $i$, we have $\|x_{t,k}^i - x_t\|^2 \leq \gamma^2 L^{-2} \|\nabla F_w(x_t)\|^2 + 4\gamma^2 L^{-2} G^2, \forall k = 0, \ldots, K$, when $\alpha \leq \min\left\{\frac{\gamma}{8KL}, \frac{\gamma}{8KLq_i^t}\right\}$ and $\gamma \leq 1/3$.*

*Proof.* During the $t$-th communication round, $S_t$ and $q_t^i$ are fixed. Then, for any $\beta > 0$ and $\alpha \leq \min\{\frac{\gamma}{\beta L}, \frac{\gamma}{\beta L q_t^i}\}$, using Lemma 6 gives

$$\|x_{k+1}^i - x_t\|^2 \leq (1 + \beta^{-1})\|x_k^i - x_t\|^2 + (1 + \beta)(\alpha)^2 (q_t^i)^2 \|\nabla f_i(x_k^i)\|^2$$

$$\leq (1 + \beta^{-1})\|x_k^i - x_t\|^2 + 3(1 + \beta)(\alpha)^2 (q_t^i)^2 \left(\|\nabla f_i(x_k^i) - \nabla f_i(x_t)\|^2\right.$$
$$+ \|\nabla f_i(x_t) - \nabla F_w(x_t)\|^2 + \|\nabla F_w(x_t)\|^2\right)$$

$$\leq (1 + \beta^{-1})\|x_k^i - x_t\|^2 + 3(1 + \beta)(\alpha q_t^{s_t})^2 \left(L^2 \|x_k^i - x_t\|^2 + 4G^2 + \|\nabla F_w(x_t)\|^2\right)$$

$$\leq (1 + \beta^{-1})\|x_k^i - x_t\|^2 + \frac{3(1 + \beta)\gamma^2}{\beta^2 L^2} \left(L^2 \|x_k^i - x_t\|^2 + 4G^2 + \|\nabla F_w(x_t)\|^2\right)$$

$$= (1 + (1 + 3\gamma^2)\beta^{-1} + 3\gamma^2 \beta^{-2})\|x_k^i - x_t\|^2 + \frac{3(1 + \beta)\gamma^2}{\beta^2 L^2} \left(4G^2 + \|\nabla F_w(x_t)\|^2\right)$$

$$\leq \exp\left(\frac{1 + 6\gamma^2}{\beta}\right)\|x_k^i - x_t\|^2 + \frac{3(1 + \beta)\gamma^2}{\beta^2 L^2} \left(4G^2 + \|\nabla F_w(x_t)\|^2\right)$$

for any $\beta \geq 1$. Unrolling the above gives for any $k = 0, \ldots, K - 1$

$$\|x_k^i - x_t\|^2 \leq \sum_{k=0}^{K-1} \exp\left(\frac{1 + 6\gamma^2}{\beta} k\right) \frac{3(1+\beta)\gamma^2}{\beta^2 L^2} \left(G^2 + \|\nabla F_w(x_t)\|^2\right)$$

which further indicates by choosing $\gamma \leq 1/3$

$$\begin{aligned}
\|x_k^i - x_t\|^2 &\leq \sum_{k=0}^{K-1} e^{2k\beta^{-1}} \frac{3(1+\beta)\gamma^2}{\beta^2 L^2} \left(4G^2 + \|\nabla F_w(x_t)\|^2\right) \\
&= \frac{1 - e^{2K/\beta}}{1 - e^{2/\beta}} \cdot \frac{3(1+\beta)\gamma^2}{\beta^2 L^2} \left(4G^2 + \|\nabla F_w(x_t)\|^2\right) \\
&\leq \frac{(e^{2K/\beta} - 1)3\gamma^2}{L^2} \left(4G^2 + \|\nabla F_w(x_t)\|^2\right) \\
&\leq \frac{\gamma^2}{L^2} \left(4G^2 + \|\nabla F_w(x_t)\|^2\right)
\end{aligned}$$

when choosing $\beta = 8K$.

$\square$

**Lemma 8.** *For any $t \geq \tau$, we have $\|x_t - x_{t-\tau}\|^2 \leq 4\gamma^2 L^{-2}\tau^2 G^2 + \gamma^2 L^{-2}\tau \sum_{l=t-\tau}^{t-1} \|\nabla F_w(x_l)\|^2$ when $\alpha \leq \min\{\frac{\gamma}{8KLq}, \frac{\gamma}{8KLq_t^i}\}$ and $\gamma \leq 1/3$.*

*Proof.* Note that

$$\begin{aligned}
\|x_{t+1} - x_t\|^2 &= \|\frac{1}{|S_t|} \sum_{i \in S_t} x_{t,K}^i - x_t\|^2 \\
&\leq \frac{1}{|S_t|} \sum_{i \in S_t} \|x_{t,K}^i - x_t\|^2 \\
&\leq \frac{\gamma^2}{L^2}(4G^2 + \|\nabla F_w(x_t)\|^2).
\end{aligned}$$

Then,

$$\begin{aligned}
\|x_t - x_{t-\tau}\|^2 &= \|\sum_{l=t-\tau}^{t-1} x_{l+1} - x_l\|^2 \\
&\leq \tau \sum_{l=t-\tau}^{t-1} \|x_{l+1} - x_l\|^2 \\
&\leq \frac{4\gamma^2}{L^2}\tau^2 G^2 + \frac{\gamma^2}{L^2}\tau \sum_{l=t-\tau}^{t-1} \|\nabla F_w(x_l)\|^2.
\end{aligned}$$

$\square$

**Lemma 9.** *For any $l \in [t-\tau, t]$ with $t \geq \tau \geq 1$ and $\alpha \leq \min\{\frac{\gamma}{8KLq}, \frac{\gamma}{8KLq_t^i}\}$ with $\gamma \leq \min\{\frac{1}{2\tau}, \frac{1}{3}\}$ we have*

$$\max_{t-\tau \leq l \leq t} \mathbb{E}\|\nabla F_w(x_l)\|^2 \leq 4\mathbb{E}\|\nabla F_w(x_{t-\tau})\|^2 + 16\tau^2\gamma^2 G^2.$$

*Proof.* For any $t - \tau \leq l \leq t$, we have

$$\begin{aligned}
\mathbb{E}\|\nabla F_w(x_l)\|^2 &\leq 2\mathbb{E}\|\nabla F_w(x_{t-\tau})\|^2 + 2\mathbb{E}\|\nabla F_w(x_l) - \nabla F_w(x_{t-\tau})\|^2 \\
&\leq 2\tau\gamma^2 \sum_{l=t-\tau}^{t-1} \mathbb{E}\|\nabla F_w(x_l)\|^2 + 8\tau^2\gamma^2 G^2 + 2\mathbb{E}\|\nabla F_w(x_{t-\tau})\|^2 \\
&\leq 2\tau^2\gamma^2 \max_{t-\tau \leq l \leq t} \mathbb{E}\|\nabla F_w(x_l)\|^2 + 8\tau^2\gamma^2 G^2 + 2\mathbb{E}\|\nabla F_w(x_{t-\tau})\|^2 \\
&\leq \frac{1}{2} \max_{t-\tau \leq l \leq t} \mathbb{E}\|\nabla F_w(x_l)\|^2 + 8\tau^2\gamma^2 G^2 + 2\mathbb{E}\|\nabla F_w(x_{t-\tau})\|^2
\end{aligned}$$

where the second inequality follows Lemma 8 and we use $\gamma \leq 1/(2\tau)$ in the last inequality. Finally, taking the maximum over $l$ on the left-hand side completes the proof. $\square$

**Lemma 10.** *Define* $F_w := \sum_{i=1}^{N} w_i f_i$ *for* $\sum_{i=1}^{N} w_i = 1, w_i \geq 0$. *Suppose Assumptions 1,2 hold. Considering any sequence* $y_t^i$ *that satisfies* $\sum_{i=1}^{N} y_t^i = 1, y_t^i \geq a^{-1} > 0, \forall i \in [N], t \geq 0$ *and letting* $q_t^i = \frac{w_i}{y_t^i}, \forall i \in [N]$, *then, given* $\tau \geq \tau_{mix} \log(1/\delta)$ *with* $0 < \delta < 1$, *for* $\alpha \leq \frac{\gamma}{8\bar{a}KL}$ *with* $\gamma \leq \min\{\frac{1}{384\tau L}, \frac{L}{384\tau}, \frac{1}{3}\}$, *we have* $\forall T > \tau$,

$$\frac{1}{T-\tau} \sum_{t=\tau}^{T-1} \mathbb{E}\|\nabla F_w(x_{t-\tau})\|^2 \leq \frac{32\bar{a}L\Delta_\tau}{\gamma(T-\tau)} + \frac{8}{T-\tau} \sum_{t=\tau}^{T-1} \mathbb{E}\left[\|\tilde{q}_t\|_\infty^2 \|\nabla F_w(x_{t-\tau})\|^2\right]$$

$$+ 32\bar{a}LG^2 \left(3\gamma + 6\gamma\tau^2 + \frac{2\gamma}{L} + \frac{3\gamma}{16L^2} + \frac{\gamma^2}{16aL}\right)$$

$$+ \frac{32G^2}{T-\tau} \sum_{t=\tau}^{T-1} \mathbb{E}\|\tilde{q}_t\|_\infty^2 + 8c_1^2\delta^2 G^2$$

*where* $\bar{a} = a \max_i\{w_i\}$, $\tilde{q}_t = (\tilde{q}_t^1, \dots, \tilde{q}_t^N)$ *with* $\tilde{q}_t^i = q_t^i - \frac{w_i}{\pi_i}$, *and* $c_1$ *is some constant. Moreover,*

$$\Delta_\tau := \mathbb{E}[F_w(x_\tau) - \min_x F_w(x)] \leq \frac{\gamma\tau}{2\bar{a}L}G^2 + \mathbb{E}[F_w(x_0) - F_w^*].$$

*Proof.* For notation simplicity, we drop subscript $t$ for $x_{t,k}^i$. Define $q_t^i = \frac{w_i}{y_t^i}$. Note that

$$x_K^i = x_t - \sum_{k=0}^{K-1} \alpha q_t^i \nabla f_i(x_k^i)$$

$$x_{t+1} = x_t - \frac{1}{B} \sum_{i \in S_t} \sum_{k=0}^{K-1} \alpha q_t^i \nabla f_i(x_k^i)$$

where $S_t$ denotes the subset of clients drawn in the $t$-th round. Due to the smoothness of every $f_i$, we have

$$\mathbb{E}[F_w(x_{t+1}) - F_w(x_t)] \leq \mathbb{E}\langle \nabla F_w(x_t), x_{t+1} - x_t \rangle + \frac{L}{2}\mathbb{E}\|x_{t+1} - x_t\|^2.$$

Considering $t \geq \tau$ for any $\tau \geq 0$,

$$\mathbb{E}\langle \nabla F_w(x_t), x_{t+1} - x_t \rangle = -\mathbb{E}\langle \nabla F_w(x_t), \frac{1}{B} \sum_{i \in S_t} \sum_{k=0}^{K-1} \alpha q_t^i \nabla f_i(x_k^i) \rangle$$

$$= \underbrace{\mathbb{E}\langle \nabla F_w(x_{t-\tau}) - \nabla F_w(x_t), \frac{1}{B} \sum_{i \in S_t} \sum_{k=0}^{K-1} \alpha q_t^i \nabla f_i(x_k^i) \rangle}_{e_1}$$

$$+ \underbrace{\mathbb{E}\langle -\nabla F_w(x_{t-\tau}), \frac{1}{B} \sum_{i \in S_t} \sum_{k=0}^{K-1} \alpha q_t^i \nabla f_i(x_{t-\tau}) \rangle}_{e_2}$$

$$+ \underbrace{\mathbb{E}\langle -\nabla F_w(x_{t-\tau}), \frac{1}{B} \sum_{i \in S_t} \sum_{k=0}^{K-1} \alpha q_t^i (\nabla f_i(x_k^i) - \nabla f_i(x_t)) \rangle}_{e_3}$$

$$+ \underbrace{\mathbb{E}\langle -\nabla F_w(x_{t-\tau}), \frac{1}{B} \sum_{i \in S_t} \sum_{k=0}^{K-1} \alpha q_t^i (\nabla f_i(x_t) - \nabla f_i(x_{t-\tau})) \rangle}_{e_4}.$$

We first note that according to the conditions on $y_t^i$, $w_i \leq q_t^i \leq aw_i$ with some positive constant $a < \infty$ for every $i \in [N]$ and $\forall t \geq 0$. Then by choosing $\alpha \leq \frac{\gamma}{8aKLw_m} \leq \min\{\frac{\gamma}{8KL}, \frac{\gamma}{8KL\max_i\{q_t^i\}}\}$ with $\gamma \leq 1/3$ and $w_m = \max_i w_i$.

$$
\begin{aligned}
e_1 &\leq \frac{1}{2}\mathbb{E}\|\nabla F_w(x_t) - \nabla F(x_{t-\tau})\|^2 + \frac{1}{2}\mathbb{E}\left\|\frac{1}{B}\sum_{i\in S_t}\sum_{k=0}^{K-1}\alpha q_t^i \nabla f_i(x_k^i)\right\|^2 \\
&\leq \frac{L^2}{2}\mathbb{E}\|x_t - x_{t-\tau}\|^2 + \mathbb{E}\left\|\frac{1}{B}\sum_{i\in S_t}\sum_{k=0}^{K-1}\alpha q_t^i(\nabla f_i(x_k^i) - \nabla f_i(x_t))\right\|^2 \\
&\quad + \mathbb{E}\left\|\frac{1}{B}\sum_{i\in S_t}\sum_{k=0}^{K-1}\alpha q_t^i \nabla f_i(x_t)\right\|^2 \\
&\leq \frac{L^2}{2}\mathbb{E}\|x_t - x_{t-\tau}\|^2 + K\mathbb{E}\left[\frac{1}{B}\sum_{i\in S_t}\sum_{k=0}^{K-1}(\alpha)^2 L^2(q_t^i)^2\|x_k^i - x_t\|^2\right] \\
&\quad + \mathbb{E}\left\|\frac{1}{B}\sum_{i\in S_t}\sum_{k=0}^{K-1}\alpha q_t^i \nabla f_i(x_t)\right\|^2 \\
&\leq \frac{\tau\gamma^2}{2}\sum_{l=t-\tau}^{t-1}\mathbb{E}\|\nabla F_w(x_l)\|^2 + 2\tau^2\gamma^2 G^2 + \frac{\gamma^2}{64L^2}\mathbb{E}\|\nabla F_w(x_t)\|^2 + \frac{\gamma^2 G^2}{16L^2} \\
&\quad + \frac{\gamma^2}{64L^2}\mathbb{E}\left\|\frac{1}{B}\sum_{i\in S_t}\nabla f_i(x_t)\right\|^2 \\
&\leq \frac{\tau\gamma^2}{2}\sum_{l=t-\tau}^{t-1}\mathbb{E}\|\nabla F_w(x_l)\|^2 + \left(2\tau^2 + \frac{1}{16L^2} + \frac{1}{8BL^2}\right)\gamma^2 G^2 + \frac{3\gamma^2}{64L^2}\mathbb{E}\|\nabla F(x_t)\|^2
\end{aligned}
$$

where we use Lemmas 7 and 8 in the fourth inequality; we use the fact $\mathbb{E}\|\frac{1}{B}\sum_{i\in S_t}\nabla f_i(x_t)\|^2 \leq 2\mathbb{E}\|\nabla F_w(x_t)\|^2 + 8G^2/B$ in the last inequality. Next we turn to bound $e_2$. Note that

$$
\begin{aligned}
e_2 &= -\alpha K\mathbb{E}\left[\mathbb{E}\left(\langle\nabla F_w(x_{t-\tau}), \frac{1}{B}\sum_{i\in S_t}q_t^i\nabla f_i(x_{t-\tau})\rangle \mid \mathcal{F}_{t-\tau}\right)\right] \\
&= \frac{\alpha K}{2}\mathbb{E}\|\nabla F_w(x_{t-\tau}) - \mathbb{E}(\frac{1}{B}\sum_{i\in S_t}q_t^i\nabla f_i(x_{t-\tau}) \mid \mathcal{F}_{t-\tau})\|^2 - \frac{\alpha K}{2}\mathbb{E}\|\nabla F_w(x_{t-\tau})\|^2 \\
&\quad - \frac{\alpha K}{2}\mathbb{E}\|\mathbb{E}(\frac{1}{B}\sum_{i\in S_t}q_t^i\nabla f_i(x_{t-\tau}) \mid \mathcal{F}_{t-\tau})\|^2 \\
&\leq \frac{\alpha K}{2}\mathbb{E}\|\nabla F_w(x_{t-\tau}) - \mathbb{E}(\frac{1}{B}\sum_{i\in S_t}q_t^i\nabla f_i(x_{t-\tau}) \mid \mathcal{F}_{t-\tau})\|^2 - \frac{\alpha K}{2}\mathbb{E}\|\nabla F_w(x_{t-\tau})\|^2 \\
&\leq \alpha K\mathbb{E}\|\nabla F_w(x_{t-\tau}) - \mathbb{E}(\frac{1}{B}\sum_{i\in S_t}q_*^i\nabla f_i(x_{t-\tau}) \mid \mathcal{F}_{t-\tau})\|^2 - \frac{\alpha K}{2}\mathbb{E}\|\nabla F_w(x_{t-\tau})\|^2 \\
&\quad + \alpha K\mathbb{E}\left\|\frac{1}{B}\sum_{i\in S_t}(q_t^i - q_*^i)\nabla f_i(x_{t-\tau})\right\|^2
\end{aligned}
$$

where $q_*^i = \frac{w_i}{\pi_i}$ and $\mathcal{F}_{t-\tau}$ is the filtration up to $t-\tau$. Next, we provide the bound for $\mathbb{E}\|\nabla F_w(x_{t-\tau}) - \mathbb{E}(\frac{1}{B}\sum_{i\in S_t}q_*^i\nabla f_i(x_{t-\tau}) \mid \mathcal{F}_{t-\tau})\|^2$. Since we are focusing on the case for a particular $R$, without causing confusion, we drop $R$ for notation simplicity in the following analysis.

Denoting $\psi_S := \lim_{t\to\infty} P(S_t = S)$, we have

$$
\pi^i = \frac{\sum_{\hat{S}_i}\psi_{\hat{S}_i}}{\sum_{i=1}^N \sum_{\hat{S}_i}\psi_{\hat{S}_i}} = \frac{\sum_{\hat{S}_i}\psi_{\hat{S}_i}}{B}
$$

where $\hat{S}_i$ denotes any set with size $B$ containing $i$. Then, for any vectors $\{v_i\}_{i=1}^N$, we have

$$\sum_{S \in \mathcal{S}} \sum_{i \in S} \frac{\psi_S}{\pi^i} v_i = \sum_{i=1}^N \sum_{\hat{S}_i} \frac{\psi_{\hat{S}_i}}{\pi^i} v_i = B \sum_{i=1}^N v_i.$$

where $\mathcal{S}$ is the collection of all sets with size $B$. Thus, by letting $v_i = w_i \nabla f_i(x_{t-\tau})$ in the above, we obtain

$$\mathbb{E}\|\nabla F_w(x_{t-\tau}) - \mathbb{E}(\frac{1}{B} \sum_{i \in S_t} q_*^i \nabla f_i(x_{t-\tau})|\mathcal{F}_{t-\tau})\|^2$$

$$= \mathbb{E}\|\nabla F_w(x_{t-\tau}) - \frac{1}{B} \sum_{S \in \mathcal{S}} \sum_{i \in S} P(S_t = S|\mathcal{F}_{t-\tau}) q_*^i \nabla f_i(x_{t-\tau})\|^2$$

$$= \mathbb{E}\left\|\frac{1}{B} \sum_{S \in \mathcal{S}} \sum_{i \in S} (P(S_t = S|\mathcal{F}_{t-\tau}) - \psi_S) q_*^i \nabla f_i(x_{t-\tau})\right\|^2$$

by noting $q_*^i = w_i/\pi^i$. Moreover, $P(S_t = \cdot)$ can be uniquely induced by $\phi_R(t)$ defined by equation 6 under proper linear transformations, which also indicates that $P(S_t = \cdot \mid \mathcal{F}_{t-\tau}) = P(S_t = \cdot \mid S_{t-\tau})$. Thus, Lemma 4 implies $|P(S_t = S \mid \mathcal{F}_{t-\tau}) - \psi_S| \leq c_1 \delta \pi_{min}/\sqrt{C_N^B}$ for some $c_1 > 0$, $\forall S$ when $\tau \geq \tau_{mix} \log(1/\delta)$ with $C_N^B = \begin{pmatrix} N \\ B \end{pmatrix}$. Then,

$$\mathbb{E}\left\|\nabla F_w(x_{t-\tau}) - \mathbb{E}(\frac{1}{B} \sum_{i \in S_t} q_*^i \nabla f_i(x_{t-\tau})|\mathcal{F}_{t-\tau})\right\|^2$$

$$= \mathbb{E}\left\|\frac{1}{B} \sum_{S \in \mathcal{S}} \sum_{i \in S} (P(S_t = S|\mathcal{F}_{t-\tau}) - \psi_S) q_*^i \nabla f_i(x_{t-\tau})\right\|^2$$

$$\leq \frac{1}{B} \mathbb{E}\left[\sum_{i \in S} \left\|\sum_{S \in \mathcal{S}} (P(S_t = S|\mathcal{F}_{t-\tau}) - \phi_S) q_*^i \nabla f_i(x_{t-\tau})\right\|^2\right]$$

$$\leq c_1^2 \pi_{min}^2 \delta^2 \mathbb{E}\left[\frac{1}{B} \sum_{i \in S} \left\|q_*^i \nabla f_i(x_{t-\tau})\right\|^2\right]$$

$$\leq c_1^2 \delta^2 (\mathbb{E}\|\nabla F_w(x_{t-\tau})\|^2 + 4G^2)$$

where we use the fact that

$$\|\nabla f_i(x_{t-\tau})\|^2 \leq 2\|\nabla F_w(x_{t-\tau})\|^2 + 8G^2.$$

Utilizing the following

$$\mathbb{E}\|\frac{1}{B} \sum_{i \in S_t} (q_t^i - q_*^i) \nabla f_i(x_{t-\tau})\|^2 = \mathbb{E}\|\frac{1}{B} \sum_{i \in S_t} \tilde{q}_t^i (\nabla f_i(x_{t-\tau}) - \nabla F(x_{t-\tau}) + \nabla F(x_{t-\tau}))\|^2$$

$$\leq 8G^2 \mathbb{E}\|\tilde{q}_t\|_\infty^2 + 2\mathbb{E}\left[\|\tilde{q}_t\|_\infty^2 \|\nabla F(x_{t-\tau})\|^2\right]$$

where we denote $\tilde{q}_t^i = q_t^i - q_*^i$. Then we bound $e_2$ as

$$e_2 \leq \frac{\alpha K}{2}(2c_1^2\delta^2 - 1)\mathbb{E}\|\nabla F(x_{t-\tau})\|^2 + 2\alpha K G^2(\delta^2 + 4\mathbb{E}\|\tilde{q}_t\|_\infty^2) + 2\alpha K \mathbb{E}\left[\|\tilde{q}_t\|_\infty^2 \|\nabla F(x_{t-\tau})\|^2\right].$$

In order to bound $e_3$, note that according to Lemma 7 for $\alpha \le \frac{\gamma}{8KLa} \le \frac{\gamma}{8KLq_t^i}$

$$
\begin{aligned}
e_3 &\le \mathbb{E}\left[\frac{1}{B}\sum_{i\in S_t}\sum_{k=0}^{K-1}\alpha q_t^i\|\nabla F(x_{t-\tau})\|\left\|\nabla f_i(x_k^i)-\nabla f_i(x_t)\right\|\right]\\
&\le \mathbb{E}\left[\frac{1}{B}\sum_{i\in S_t}\sum_{k=0}^{K-1}\left(\frac{(\alpha q_t^i)^2 K}{2}\|\nabla F_w(x_{t-\tau})\|^2+\frac{L^2}{2K}\|x_k^i-x_t\|^2\right)\right]\\
&\le \mathbb{E}\left[\frac{1}{B}\sum_{i\in S_t}\sum_{k=0}^{K-1}\left(\frac{\gamma^2}{128L^2K}\|\nabla F_w(x_{t-\tau})\|^2+\frac{\gamma^2}{2K}(\|\nabla F_w(x_t)\|^2+4G^2)\right)\right]\\
&\le \frac{\gamma^2}{128L^2}\mathbb{E}\|\nabla F_w(x_{t-\tau})\|^2+\frac{\gamma^2}{2}\mathbb{E}\|\nabla F_w(x_t)\|^2+2\gamma^2 G^2.
\end{aligned}
$$

Finally, based on Lemma 8, similarly we obtain

$$
\begin{aligned}
e_4 &\le \mathbb{E}\left[\frac{1}{B}\sum_{i\in S_t}\sum_{k=0}^{K-1}\alpha q_t^i\|\nabla F(x_{t-\tau})\|\|\nabla f_i(x_t)-\nabla f_i(x_{t-\tau})\|\right]\\
&\le \mathbb{E}\left[\frac{1}{B}\sum_{i\in S_t}\sum_{k=0}^{K-1}\left(\frac{(\alpha q_t^i)^2 K}{2}\|\nabla F(x_{t-\tau})\|^2+\frac{L^2}{2K}\|x_t-x_{t-\tau}\|^2\right)\right]\\
&\le \frac{\gamma^2}{128L^2}\mathbb{E}\|\nabla F(x_{t-\tau})\|^2+\frac{\gamma^4\tau}{2}\left(\sum_{l=t-\tau}^{t-1}\mathbb{E}\|\nabla F_w(x_l)\|^2+4\tau G^2\right).
\end{aligned}
$$

Thus, denoting $\bar{a}=a\max_i\{w_i\}$

$$
\begin{aligned}
\frac{\gamma}{16\bar{a}L}\mathbb{E}\|\nabla F_w(x_{t-\tau})\|^2 &\le \mathbb{E}[F_w(x_t)-F_w(x_{t+1})]+\frac{\tau\gamma^2(1+\gamma^2)}{2}\sum_{l=t-\tau}^{t-1}\mathbb{E}\|\nabla F_w(x_l)\|^2\\
&\quad+\left(\frac{\gamma^2}{2}+\frac{\gamma^2}{2L}+\frac{3\gamma^2}{64L^2}\right)\mathbb{E}\|\nabla F_w(x_t)\|^2+\frac{\gamma}{aL}G^2\mathbb{E}\|\tilde{q}_t\|_\infty^2\\
&\quad+\frac{\gamma}{4\bar{a}L}\mathbb{E}\left[\|\tilde{q}_t\|_\infty^2\|\nabla F_w(x_{t-\tau})\|^2\right]+\frac{\gamma c_1^2\delta^2}{4\bar{a}L}G^2\\
&\quad+\gamma^2 G^2\left(2+2\tau^2+2\gamma^2\tau^2+\frac{2}{L}+\frac{3}{16L^2}\right)\\
&\quad+\left(\frac{\gamma\delta^2}{8\bar{a}L}+\frac{\gamma^2}{64L^2}\right)\mathbb{E}\|\nabla F_w(x_{t-\tau})\|^2
\end{aligned}
$$

which implies that

$$
\begin{aligned}
\gamma\mathbb{E}\|\nabla F_w(x_{t-\tau})\|^2 \leq{} & 16\bar{a}L\mathbb{E}[F_w(x_t) - F_w(x_{t+1})] + \frac{16\bar{a}L\tau\gamma^2(1+\gamma^2)}{2}\sum_{l=t-\tau}^{t-1}\mathbb{E}\|\nabla F_w(x_l)\|^2 \\
& + \gamma\left(16\bar{a}L\gamma + 8\bar{a}\gamma + \frac{3\bar{a}\gamma}{4L}\right)\mathbb{E}\|\nabla F_w(x_t)\|^2 + 4\gamma c_1^2\delta^2 G^2 \\
& + 4\gamma\mathbb{E}\left[\|\tilde{q}_t\|_\infty^2\|\nabla F_w(x_{t-\tau})\|^2\right] + 16\gamma G^2\mathbb{E}\|\tilde{q}_t\|_\infty^2 \\
& + 16\bar{a}L\gamma^2 G^2\left(2 + 2\tau^2 + 2\gamma^2\tau^2 + \frac{2}{L} + \frac{3}{16L^2}\right) \\
& + \gamma\left(2c_1^2\delta^2 + \frac{\bar{a}\gamma}{4L}\right)\mathbb{E}\|\nabla F_w(x_{t-\tau})\|^2 \\
\leq{} & 16\bar{a}L\mathbb{E}[F_w(x_t) - F_w(x_{t+1})] + \gamma\left(16\bar{a}L\gamma + 8\bar{a}\gamma + \frac{3\bar{a}\gamma}{4L}\right)\mathbb{E}\|\nabla F_w(x_t)\|^2 \\
& + \gamma\left(2c_1^2\delta^2 + \frac{\bar{a}\gamma}{4L} + 32\bar{a}L\tau\gamma(1+\gamma^2)\right)\mathbb{E}\|\nabla F_w(x_{t-\tau})\|^2 \\
& + 4\gamma\mathbb{E}\left[\|\tilde{q}_t\|_\infty^2\|\nabla F_w(x_{t-\tau})\|^2\right] + 16\gamma G^2\mathbb{E}\|\tilde{q}_t\|_\infty^2 + 4\gamma c_1^2\delta^2 G^2 \\
& + 16\bar{a}L\gamma^2 G^2\left(2 + 2\tau^2 + 2\gamma^2\tau^2 + 8\tau^4\gamma^2(1+\gamma^2) + \frac{2}{L} + \frac{3}{16L^2}\right)
\end{aligned}
$$

where we make use of

$$
\sum_{l=t-\tau}^{t-1}\mathbb{E}\|\nabla F_w(x_l)\|^2 \leq 4\tau\mathbb{E}\|\nabla F_w(x_{t-\tau})\|^2 + 16\tau^3\gamma^2 G^2
$$

by Lemma 9. Under the following conditions

$$
2c_1^2\delta^2 \leq \frac{1}{6}, \quad \frac{\bar{a}\gamma}{4L} \leq \frac{1}{36}, \quad \gamma \leq \min\{\frac{1}{2\tau}, \frac{1}{384\bar{a}}\},
$$

$$
64\bar{a}L\tau\gamma \leq \frac{1}{12},
$$

which implies $32aL\tau\gamma(1+\gamma^2) \leq \frac{1}{6}$ and hence $2c_1^2\delta^2 + \frac{\bar{a}\gamma}{4L} + 32\bar{a}L\tau\gamma(1+\gamma^2) \leq \frac{1}{2}$, then we obtain

$$
\begin{aligned}
\gamma\mathbb{E}\|\nabla F_w(x_{t-\tau})\|^2 \leq{} & 32\bar{a}L\mathbb{E}[F_w(x_t) - F_w(x_{t+1})] + 2\gamma\left(16\bar{a}L\gamma + 8\bar{a}\gamma + \frac{3\bar{a}\gamma}{4L}\right)\mathbb{E}\|\nabla F_w(x_t)\|^2 \\
& + 8\gamma\mathbb{E}\left[\|\tilde{q}_t\|_\infty^2\|\nabla F_w(x_{t-\tau})\|^2\right] + 32\gamma G^2\mathbb{E}\|\tilde{q}_t\|_\infty^2 + 8\gamma c_1^2\delta^2 G^2 \\
& + 32\bar{a}L\gamma^2 G^2\left(2 + 2\tau^2 + 2\gamma^2\tau^2 + 8\tau^4\gamma^2(1+\gamma^2) + \frac{2}{L} + \frac{3}{16L^2}\right).
\end{aligned}
$$

Summing over $\tau \leq t \leq T-1$ gives

$$
\begin{aligned}
\gamma\sum_{t=\tau}^{T-1}\mathbb{E}\|\nabla F_w(x_{t-\tau})\|^2 \leq{} & 32\bar{a}L\Delta_\tau + 2\gamma\left(16\bar{a}L\gamma + 8\bar{a}\gamma + \frac{3\bar{a}\gamma}{4L}\right)\sum_{t=\tau}^{T-1}\mathbb{E}\|\nabla F_w(x_t)\|^2 \\
& + 8\gamma\sum_{t=\tau}^{T-1}\mathbb{E}\left[\|\tilde{q}_t\|_\infty^2\|\nabla F_w(x_{t-\tau})\|^2\right] + 32\gamma G^2\sum_{t=\tau}^{T-1}\mathbb{E}\|\tilde{q}_t\|_\infty^2 \\
& + 32\bar{a}L\gamma^2 G^2\left(3 + 6\tau^2 + \frac{2}{L} + \frac{3}{16L^2}\right)(T-\tau) + 8\gamma c_1^2\delta^2 G^2(T-\tau).
\end{aligned}
$$

where $\Delta_\tau = \mathbb{E}[F_w(x_\tau) - F^*]$ and we use $\gamma^2\tau^2 \leq 1/4$. Again leveraging Lemma 9, we observe

$$
\sum_{t=\tau}^{T-1}\mathbb{E}\|\nabla F_w(x_t)\|^2 \leq 4\sum_{t=\tau}^{T-1}\mathbb{E}\|\nabla F_w(x_{t-\tau})\|^2 + 16\tau^2\gamma^2 G^2(T-\tau)
$$

which thus renders

$$\frac{1}{T-\tau}\sum_{t=\tau}^{T-1}\mathbb{E}\|\nabla F_w(x_{t-\tau})\|^2 \le \frac{32\bar{a}L\Delta_\tau}{\gamma(T-\tau)} + \frac{8}{T-\tau}\sum_{t=\tau}^{T-1}\mathbb{E}\left[\|\tilde{q}_t\|_\infty^2\|\nabla F_w(x_{t-\tau})\|^2\right]$$

$$+ 32\bar{a}LG^2\left(3\gamma + 6\gamma\tau^2 + \frac{2\gamma}{L} + \frac{3\gamma}{16L^2} + \frac{\gamma^2}{16aL}\right) + 8c_1^2\delta^2G^2$$

$$+ \frac{32G^2}{T-\tau}\sum_{t=\tau}^{T-1}\mathbb{E}\|\tilde{q}_t\|_\infty^2$$

by noting that $16\bar{a}L\gamma + 8a\gamma + \frac{3\bar{a}\gamma}{4L} \le \frac{1}{16}$.

In the following, we turn to bound $\Delta_\tau$. Noting that

$$F_w(x_{t+1}) - F_w(x_t) \le -\alpha K\langle\nabla F_w(x_t), \frac{1}{BK}\sum_{i\in S_t}\sum_{k=0}^{K-1}\nabla f_i(x_k^i)\rangle + \frac{\alpha^2 L}{2B^2}\left\|\sum_{i\in S_t}\sum_{k=0}^{K-1}\nabla f_i(x_k^i)\right\|^2$$

$$\le \frac{\alpha K}{2}\left\|\frac{1}{BK}\sum_{i\in S_t}\sum_{k=0}^{K-1}(\nabla f_i(x_k^i) - \nabla F_w(x_t))\right\|^2 - \frac{\alpha K}{2}\|\nabla F_w(x_t)\|^2$$

by $\alpha \le \frac{\gamma}{8aLK} \le \frac{1}{2LK}$. Moreover, since

$$\left\|\frac{1}{BK}\sum_{i\in S_t}\sum_{k=0}^{K-1}(\nabla f_i(x_k^i) - \nabla F_w(x_t))\right\|^2 \le \frac{2}{BK}\sum_{i\in S_t}\sum_{k=0}^{K}(L^2\|x_k^i - x_t\|^2 + 4G^2)$$

$$\le 2\gamma^2\|\nabla F_w(x_t)\|^2 + 8G^2$$

we conclude that

$$F_w(x_{t+1}) - F_w(x_t) \le -\frac{\alpha K}{2}(1-2\gamma^2)\|\nabla F_w(x_t)\|^2 + 4\alpha KG^2 \le \frac{\gamma}{2\bar{a}L}G^2$$

which implies

$$\Delta_\tau = \mathbb{E}[F_w(x_\tau) - F^*] \le \frac{\gamma\tau}{2\bar{a}L}G^2 + F_w(x_0) - F_w^*.$$

$\square$

## F CONVERGENCE ANALYSIS OF FEDAVG UNDER CORRELATED CLIENT PARTICIPATION

In this section, we provide the convergence analysis of Vanilla FedAvg for correlated client participation. We first show FedAvg suffers from unavoidable bias (stated in Theorem 1), summarized by the following proposition.

**Proposition 3.** *There exists a problem case such that FedAvg converges with unavoidable asymptotic bias.*

*Proof.* We consider a problem case with $N = 3, B = 1, R = 1$. We set $p_1 = 0.25, p_2 = 0.25, p_3 = 0.5$ and $f_i(x) = \frac{1}{2}(x-i)^2, i = 1, 2, 3$ and $x \in \mathbb{R}$. In this case, we have the Markov chain induced by the problem denoted by $P \in \mathbb{R}^{3\times3}$. Letting $\pi \in \mathbb{R}^3$ be the stationary distribution of $P$, a straightforward calculation gives $\pi_1 = \pi_2 = 0.3, \pi_3 = 0.4$. Then we obtain the server's update of FedAvg given by

$$x_{t+1} = \beta x_t + (1-\beta)i_t$$

where $\beta = (1-\alpha)^K < 1$ with $\alpha$ being the stepsize of local updates; $i_t$ is the index of the sampled client at round $t$ which is a random variable. Taking the expectation on both sides yields

$$\mathbb{E}[x_{t+1}] = \beta\mathbb{E}[x_t] + (1-\beta)\mu^T P^t I$$

$$= \beta\mathbb{E}[x_t] + (1-\beta)(\mu^T P^t - \pi^T)I + (1-\beta)\pi^T I$$

where $\mu = (p_1, p_2, p_3)$, and $I = (1, 2, 3)$ is the vector formed by clients' indices. Noting that the third term vanishes as $t \to \infty$ due to the convergence the Markov chain (shown by Lemma 4), we conclude that $\lim_{t \to \infty} \mathbb{E}[x_t] = \sum_{i=1}^{3} \pi_i i$ which is the minimizer of $F_\pi(x) := \sum_{i=1}^{3} \pi_i f_i(x)$ but not $F(x) = \frac{1}{3} \sum_{i=1}^{3} f_i(x)$. And $|F'(\pi^T I)| = |I^T(\pi - \frac{1}{3}\mathbf{1}_3)|$. Therefore, the bias in Theorem 1 is unavoidable. $\qquad\square$

Then we show the convergence result of FedAvg.

**Theorem 4.** *Suppose Assumptions 1,2 hold and assume $\|\nabla F(x)\| \leq D, \forall x$. Then, by choosing $\alpha = \mathcal{O}(\frac{\gamma}{K})$ and $T \geq 2\tau_{mix} \log \tau_{mix}$, the output $\tilde{x}_T$ generated by FedAvg satisfies*

$$\mathbb{E}\|\nabla F(\tilde{x}_T)\|^2 = \mathcal{O}\left(\frac{\Delta_0}{\gamma T}\right) + \mathcal{O}\left(\frac{\tau_{mix} \log T G^2}{T}\right) + \mathcal{O}\left((\gamma \tau_{mix}^2 \log^2 T + \gamma^2)G^2\right)$$
$$+ \mathcal{O}\left((G^2 + D^2)\|\pi - \frac{1}{N}\mathbf{1}_N\|_1^2\right)$$

*where $\Delta_0 := \mathbb{E}[F(x_0) - \min_x F(x)]$ and $\tau_{mix}$ is the mixing time.*

*Proof.* For FedAvg, we have $y_t^i = 1/N$. Utilizing Lemma 10 and setting $w_i = \frac{1}{N}$, it yields

$$\frac{1}{T-\tau} \sum_{t=0}^{T-\tau-1} \mathbb{E}\|\nabla F(x_t)\|^2 \leq \frac{32L\Delta_0}{\gamma(T-\tau)} + \frac{8D^2}{T-\tau} \sum_{t=\tau}^{T-1} \mathbb{E}\|\tilde{q}_t\|_\infty^2 + \frac{36G^2}{T-\tau} \sum_{t=\tau}^{T-1} \mathbb{E}\|\tilde{q}_t\|_\infty^2 + \frac{16\tau G^2}{T-\tau}$$
$$+ 32LG^2\left(3\gamma + 6\gamma\tau^2 + \frac{2\gamma}{L} + \frac{3\gamma}{16L^2} + \frac{\gamma^2}{16L}\right) + 8c_1^2\delta^2 G^2.$$

Then noting that $\|\tilde{q}_t\|_\infty^2 \leq \pi_{min}^{-2}\|\pi - \frac{1}{N}\mathbf{1}_N\|_1^2$, we conclude

$$\mathbb{E}\|\nabla F(\tilde{x}_T)\|^2 = \mathcal{O}\left(\frac{\Delta_0}{\gamma T}\right) + \mathcal{O}\left(\frac{\tau G^2}{T}\right) + \mathcal{O}\left((\gamma\tau^2 + \gamma^2)G^2\right) + \mathcal{O}\left((G^2 + D^2)\|\pi - \frac{1}{N}\mathbf{1}_N\|_1^2\right)$$

by setting $\delta = 1/\sqrt{T}$. For the above to be true, we need $T \geq \tau = \tau_{mix} \log T$, which is actually always satisfied for $T \geq 2\tau_{mix} \log \tau_{mix}$. To see this, we observe that if $T \leq \tau_{mix}^2$, $\tau_{mix} \log T \leq 2\tau_{mix} \log \tau_{mix}$; if $T \geq \tau_{mix}^2$, $\tau_{mix} \log T \leq \sqrt{T} \log T \leq T$. This completes the proof. $\qquad\square$

The following corollary restates the convergence result of Theorem 1.

**Corollary 1.** *Suppose all conditions in Theorem 4 hold. Then, choosing $\alpha = \tilde{\mathcal{O}}(1/(K\tau_{mix}\sqrt{T}))$, the output $\tilde{x}_T$ of FedAvg satisfies*

$$\mathbb{E}\|\nabla F(\tilde{x}_T)\|^2 \leq \tilde{\mathcal{O}}\left(\frac{\tau_{mix}}{\sqrt{T}}\right) + \mathcal{O}\left((D^2 + G^2)\|\pi_R - \frac{1}{N}\mathbf{1}_N\|_1^2\right).$$

*Proof.* The proof is straightforward by simply plugging in $\gamma = \mathcal{O}(1/(\tau\sqrt{T}))$ and $\tau = \tau_{mix} \log T$ to Theorem 4. $\qquad\square$

# G  CONVERGENCE ANALYSIS OF ALGORITHM 1

We first provide the following theorem showing that $y_t^i$ serves as a reasonable estimation of $\pi^i$.

**Theorem 5.** *For any real-valued function $f \in \mathbb{R}^N$ and any initial distribution $\mu \in \mathbb{R}^N$, we have the following:*

$$\mathbb{E}_\mu\left(\frac{1}{T}\sum_{t=0}^{T-1} f(X_t) - \pi_R^T f\right) = \frac{1}{T}\sum_{t=0}^{T-1} \mu^T Q_\mu^\dagger (P_R^t - \mathbf{1}\zeta_R^T)Q_R f$$

$$T\mathbb{E}_{\pi_R}\left(\frac{1}{T}\sum_{t=0}^{T-1} f(X_t) - \pi_R^T f\right)^2 \leq f^T \Pi_R(I - \mathbf{1}_N \pi_R^T)f + c_0 \pi_{max}\|f\|_\infty^2 N\tau_{mix}$$

$$T\mathbb{E}_\mu\left(\frac{1}{T}\sum_{t=0}^{T-1} f(X_t) - \pi_R^T f\right)^2 \leq T\mathbb{E}_{\pi_R}\left(\frac{1}{T}\sum_{t=0}^{T-1} f(X_t) - \pi_R^T f\right)^2 + 3c_0 N^2 \|g\|_\infty^2 \tau_{mix}$$

where $\mathbb{E}_\mu(\cdot)$ means the initial state $X_0$ follows $\mu$; $\Pi_R = \mathrm{diag}(\pi_R[i])$ and $Q^\dagger_\mu$ is defined such that $\mu^T Q^\dagger_\mu = \zeta_\mu$ and $\zeta^T_\mu Q_R = \mu$; $g = f - \pi^T_R f \mathbf{1}_N$; $\tau_{mix}$ is the mixing time of $P_R$.

*Proof.* We firstly show the first equality. Note that

$$\mathbb{E}_\mu\left(\frac{1}{T}\sum_{t=0}^{T-1} f(X_t) - \pi^T_R f\right) = \frac{1}{T}\sum_{k=0}^{T-1}(\mu^T P^k_R Q_R f - \zeta^T_R Q_R f)$$

$$= \frac{1}{T}\sum_{k=0}^{T-1} \mu^T(P^k_R - \mathbf{1}\zeta^T_R)Q_R f$$

where we observe that $\mu^T \mathbf{1} = 1$.

Then we turn to show the second inequality. By the definition, we have

$$T\mathbb{E}_{\pi_R}\left(\frac{1}{T}\sum_{t=0}^{T-1} f(X_t) - \pi^T_R f\right)^2 = \mathrm{Var}_{\pi_R}(f(X_0)) + \frac{2}{T}\sum_{k=1}^{T-1}(T-k)\mathrm{Cov}_{\pi_R}(f(X_0), f(X_k)). \tag{11}$$

For any $k$, let $\zeta_k$ and $\pi_k$ be the distributions after the Markov chain evolves $k$ steps. Then, we have $\zeta^T_{k+1} = \zeta^T_k P_R$ and $\pi^T_k = \zeta^T_k Q_R$. Defining $\hat{Q}_k \in \mathbb{R}^{N \times d(M,R)}$ as an inverse mapping from $\pi_k$ to $\zeta_k$, i.e., $\zeta^T_k = \pi^T_k \hat{Q}_k$, it is straightforward to verify that we can always pick a nonnegative $\hat{Q}_k$ such that $\hat{Q}_k \mathbf{1} = \mathbf{1}_N$ in the sense that the freedom of $\hat{Q}_k$ is $(N-1) \times d(M,R) - N$ when forcing both $\zeta^T_k = \pi^T_k \hat{Q}_k$ and $\hat{Q}_k \mathbf{1} = \mathbf{1}_N$ to hold. Moreover,

$$\mathrm{Cov}_{\pi_R}(f(X_0), f(X_k)) = \sum_i \pi_R[i] f(i) \sum_j [\hat{Q}_k P^k_R Q_R]_{i,j} f(j) - \sum_{i,j} \pi_R[i]\pi_R[j]f(i)f(j)$$

$$= f^T \Pi_R \hat{Q}_k P^k_R Q_R f - f^T \Pi_R \mathbf{1}_N \pi^T_R f$$

$$= f^T \Pi_R \hat{Q}_k (P^k_R - \mathbf{1}\zeta^T_R)Q_R f$$

where we utilize $\hat{Q}_k \mathbf{1} = \mathbf{1}_N$. Further, $\|\hat{Q}_k\|_\infty = 1, \forall k \geq 0$ since $\hat{Q}_k$ is nonnegative. Then,

$$\mathrm{Cov}_{\pi_R}(f(X_0), f(X_k)) \leq \pi_{\max}\|f\|^2_\infty \|Q_R\|_\infty \|P^k_R - \mathbf{1}\zeta^T_R\|_\infty.$$

Substituting it into equation 11 yields

$$T\mathbb{E}_{\pi_R}\left(\frac{1}{T}\sum_{t=0}^{T-1} f(X_t) - \pi^T_R f\right)^2 \leq \mathrm{Var}_{\pi_R}(f(X_0)) + 2\sum_{k=1}^{\infty}\mathrm{Cov}_{\pi_R}(f(X_0), f(X_k))$$

$$\leq \mathrm{Var}_{\pi_R}(f(X_0)) + 2\pi_{\max}\|f\|^2_\infty \|Q_R\|_\infty \sum_{k=0}^{T}\|P^k_R - \mathbf{1}\zeta^T_R\|_\infty$$

$$\leq f^T \Pi_R(I - \mathbf{1}_N \pi^T_R)f + c_0 \pi_{\max}\|f\|^2_\infty\|Q_R\|_\infty \tau_{mix}$$

where we make use of Lemma 5. Finally noting that $\|Q_R\|_\infty \leq \|Q_{R,1}\|_\infty\|Q_{R,2}\|_\infty \leq N$ completes the proof of the second inequality.

To obtain the third inequality, defining $g(i) = f(i) - \pi^T_R f$ we aim to bound

$$T\left|\mathbb{E}_\mu\left(\frac{1}{T}\sum_{k=0}^{T-1} g(X_k)\right)^2 - \mathbb{E}_{\pi_R}\left(\frac{1}{T}\sum_{k=0}^{T-1} g(X_k)\right)^2\right|$$

$$\leq \left|\frac{1}{T}\sum_{k=0}^{T-1}\mathbb{E}_\mu g^2(X_k) - \mathbb{E}_{\pi_R}g^2(X_k)\right| + \frac{2}{T}\sum_{k=0}^{T-1}\sum_{l=k+1}^{T-1}\left|\mathbb{E}_\mu(g(X_k)g(X_l)) - \mathbb{E}_{\pi_R}(g(X_k)g(X_l))\right|.$$

For notation simplicity, we drop the subscript $R$ without confusion to get

$$\left| \mathbb{E}_\mu (g(X_k)g(X_l)) - \mathbb{E}_{\pi_R}(g(X_k)g(X_l)) \right|$$

$$= \left| \sum_{i,j} \mu_i g(j) ((\hat{Q}_k P^k Q)_{i,j} - \pi_j) \sum_r ((\hat{Q}_l P^{l-k} Q)_{j,r} - \pi_r) g(r) \right|$$

$$= \left| \sum_{i,j} \mu_i g(j) (\hat{Q}_k (P^k - \mathbf{1}\zeta^T)Q)_{i,j} \sum_r (\hat{Q}_l (P^{l-k} - \mathbf{1}\zeta^T)Q)_{j,r} g(r) \right|$$

$$\leq \|g\|_\infty^2 N^2 \|P^l - \mathbf{1}\zeta^T\|_\infty.$$

Thus, by Lemma 5,

$$T \left| \mathbb{E}_\mu \left( \frac{1}{T} \sum_{k=0}^{T-1} g(X_k) \right)^2 - \mathbb{E}_{\pi_R} \left( \frac{1}{T} \sum_{k=0}^{T-1} g(X_k) \right)^2 \right|$$

$$\leq \frac{1}{T} \sum_{k=0}^{T-1} \mu^T Q_\mu^\dagger (P^k - \mathbf{1}\zeta^T) Q g^2 + \frac{2}{T} c_0 N^2 \|g\|_\infty^2 \sum_{k=0}^{T-1} \tau_{mix}$$

$$\leq \frac{1}{T} \sum_{k=0}^{T-1} \mu^T Q_\mu^\dagger (P^k - \mathbf{1}\zeta^T) Q g^2 + 2 c_0 N^2 \|g\|_\infty^2 \tau_{mix}$$

$$\leq \frac{1}{T} c_0 \|g\|_\infty^2 N \tau_{mix} + 2 c_0 N^2 \|g\|_\infty^2 \tau_{mix}$$

$$\leq 3 c_0 N^2 \|g\|_\infty^2 \tau_{mix}$$

where $g^2$ denotes the elementwise square of $g$. Combining all the above completes the proof. $\qquad\square$

Then, the following corollary induced by Theorem 5 is exactly Lemma 2.

**Corollary 2.** *Given initial $\lambda_0 = \mathbf{0}_N$ and let $\nu_t^i = \frac{1}{\lambda_t^i N}$ as in Algorithm 1, we have*

$$\mathbb{E}\|\tilde{\nu}_t\|_\infty^2 \leq \mathcal{O}\left( \frac{\tau_{mix}}{t} \right)$$

*where $\tilde{\nu}_t^i = \nu_t^i - \frac{1}{\pi_i N}$ and $\tilde{\nu}_t = (\tilde{\nu}_t^1, \dots, \tilde{\nu}_t^N)$.*

*Proof.* By Theorem 5, setting $f = \mathbf{e}_i$ for any $i$, we have

$$\mathbb{E}(\lambda_t^i - \pi_i)^2 = \mathcal{O}\left( \frac{N^2 \tau_{mix}}{t} \right) \tag{12}$$

Note that

$$\mathbb{E}(\tilde{\nu}_t^i)^2 = \frac{1}{N^2} \mathbb{E}\left( \frac{\lambda_t^i - \pi_i}{\lambda_t^i \pi_i} \right)^2 = \frac{1}{N^2} \mathbb{E}\left[ \left( \frac{\lambda_t^i - \pi_i}{\lambda_t^i \pi_i} \right)^2 \middle| \lambda_t^i \geq a \right] P(\lambda_t^i \geq a)$$

$$+ \frac{1}{N^2} \mathbb{E}\left[ \left( \frac{\lambda_t^i - \pi_i}{\lambda_t^i \pi_i} \right)^2 \middle| \lambda_t^i < a \right] P(\lambda_t^i < a) \tag{13}$$

for any positive $a$. Moreover, due to the Markov chain in Section 3 is irreducible by Lemma 1, every client will be visited infinitely as $t$ goes to infinite, which then implies there always exists some strictly positive constant $a_0$ independent of $t$ such that $\lambda_t^i \geq a_0 > 0$ almost surely for any $i \in [N]$. Combining equation 12, equation 13 we conclude

$$\mathbb{E}\|\tilde{\nu}_t^i\|_\infty^2 = \mathcal{O}\left( \frac{\tau_{mix}}{t} \right).$$

$$\square$$

### G.1 CONVERGENCE PROOF OF ALGORITHM 1

The following lemma is useful to derive the convergence proof of Algorithm 1.

**Lemma 11.** *Supposing that the stochastic scalar sequence $\mathbb{E}[U_1(t)^2] \leq u(t)$ with $u$ being a monotonically decreasing positive function w.r.t. $t$ and assuming that $U_1(t) \leq \bar{u} < \infty$ almost surely, then given any $\delta, \epsilon > 0$, for all $t \geq \inf\{t_0 \mid u(t_0)/\delta^2 \leq \epsilon/\bar{u}^2\}$ and stochastic scalar sequence $U_2(t)$,*

$$\mathbb{E}\left[U_1(t)^2 U_2(t)\right] \leq (\epsilon + \delta^2)\mathbb{E}[U_2(t)].$$

*Proof.* For any $\delta > 0$, we have for all $t \geq \inf\{t_0 \mid u(t_0)/\delta^2 \leq \epsilon/\bar{u}^2\}$

$$\begin{aligned}
\mathbb{E}[U_1(t)^2 U_2(t)] &= P(U_1(t) > \delta)\mathbb{E}[U_1(t)^2 U_2(t) \mid U_1(t) > \delta] + P(U_1(t) \leq \delta)\mathbb{E}[U_1(t)^2 U_2(t) \mid U_1(t) \leq \delta] \\
&\leq P(U_1(t) > \delta)\bar{u}^2\mathbb{E}[U_2(t)] + \delta^2\mathbb{E}[U_2(t)] \\
&\leq (\epsilon + \delta^2)\mathbb{E}[U_2(t)]
\end{aligned}$$

where we use the Markov inequality in the last step, i.e.,

$$P(U_1(t) > \delta) \leq P(U_1(t)^2 > \delta^2) \leq \frac{u(t)}{\delta^2}.$$

$\square$

Then we are ready to provide the proof for Theorem 3.

*Proof of Theorem 3:* As discussed in the proof of Corollary 2, we know that there exists a positive $a^{-1}$ which lower bounds each $\lambda_t^i$ for all $t$ almost surely, implying that $\tilde{\nu}_t^i \leq \frac{1}{N}(a + \pi_{min}^{-1})$. Then for any $t > \tau > c'\tau_{mix}$ (with $c'$ being some constant), we have

$$\mathbb{E}\left[\|\tilde{\nu}_t\|_\infty^2 \|\nabla F(x_{t-\tau})\|^2\right] \leq \frac{1}{16}\mathbb{E}\|\nabla F(x_{t-\tau})\|^2$$

by Lemmas 2 and 11. Further Utilizing Lemma 10 with setting $w_i = \frac{1}{N}$, we obtain

$$\begin{aligned}
\frac{1}{T-\tau}\sum_{t=\tau}^{T-1}\mathbb{E}\|\nabla F(x_{t-\tau})\|^2 \leq{}& \frac{64\bar{a}L\Delta_0}{\gamma(T-\tau)} + \frac{64G^2}{T-\tau}\sum_{t=\tau}^{T-1}\mathbb{E}\|\tilde{\nu}_t\|_\infty^2 + \frac{32\tau G^2}{T-\tau} + 16c_1^2\delta^2 G^2 \\
&+ 64\bar{a}LG^2\left(3\gamma + 6\gamma\tau^2 + \frac{2\gamma}{L} + \frac{3\gamma}{16L^2} + \frac{\gamma^2}{16aL}\right)
\end{aligned}$$

for $\tau \geq \tau_{mix}\max\{c', \log(1/\delta)\}$. Similar to the proofs of Theorem 4, setting $\delta = 1/\sqrt{T}$, with $T \geq c^\dagger\tau_{mix}\log\tau_{mix}$ for some constant $c^\dagger$, we finally conclude that

$$\mathbb{E}\|\nabla F(\tilde{x}_T)\|^2 = \tilde{\mathcal{O}}\left(\frac{\tau_{mix}}{\sqrt{T}}\right) + \mathcal{O}\left(\frac{1}{T}\right)$$

by choosing $\gamma = \mathcal{O}(1/(\tau\sqrt{T}))$ with $\tau = \Omega(\tau_{mix}\log T)$ and by leveraging the fact that $\sum_{t=\tau}^{T-1}\mathbb{E}\|\tilde{\nu}_t\|_\infty^2 = \mathcal{O}(\tau_{mix}\log T)$ implied by Lemma 2. This completes the proof.

## H   ADDITIONAL EXPERIMENTS

In this section, we compare Vanilla FedAvg, Debiasing FedAvg (ours) and FedVARP under the CIFAR10 dataset given the same participation pattern as in Section 6. Each client maintains a CNN with three convolution layers. Learning rates for three algorithms are selected to be with the order of $\mathcal{O}(10^{-3})$. In Figure 4, Debiasing FedAvg achieves the highest training accuracy due to its debiasing nature as shown in Theorem 3, while Vanilla FedAvg and FedVARP suffer from bias. Moreover, in Table 1, the training and test accuracies for different $R$ are presented, where one can see that Debiasing FedAvg achieves the best performance.

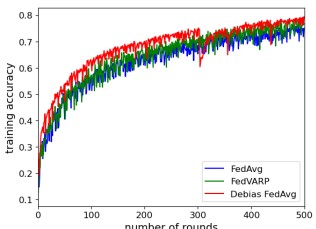
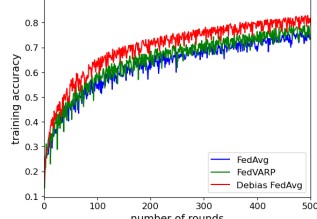
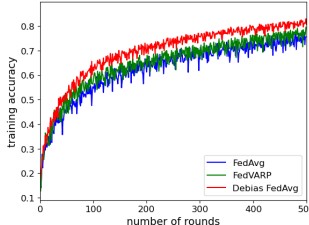

(a) FedAvg, FedVARP, Debiasing FedAvg when $R = 5$ (CIFAR10)

(b) FedAvg, FedVARP, Debiasing FedAvg when $R = 10$ (CIFAR10)

(c) FedAvg, FedVARP, Debiasing FedAvg when $R = 15$ (CIFAR10)

Figure 4: Experiments on CIFAR10. The training accuracies of Vanilla FedAvg, Debiasing FedAvg (ours) and FedVARP are compared given different values of $R$. The results show that Debiasing FedAvg achieves the highest accuracy and outperforms the other two, since FedAvg and FedVARP suffer from bias.

| Algorithms | $R = 5$ | | $R = 10$ | | $R = 15$ | |
|---|---|---|---|---|---|---|
| | Train acc | Test acc | Train acc | Test acc | Train acc | Test acc |
| FedAvg | 74.9% | 67.6% | 75.3% | 68.8% | 75.6% | 70.1% |
| FedVARP | 76.7% | 68.0% | 77.5% | 69.4% | 79.5% | 72.5% |
| Debiasing FedAvg | **79.3%** | **73.8%** | **81.5%** | **74.9%** | **82.9%** | **74.1%** |

Table 1: Training and test accuracies for different $R$ under CIFAR10

## I    THE INFLUENCE OF $R$ ON CONVERGENCE RATES

In this section, we discuss the effect of different values of $R$ on the convergence rates of Debiasing FedAvg and Vanilla FedAvg as observed empirically in Figure 3. We simulate the "effective" client sampling distribution (i.e., $\eta_R(t)$) as time evolves for different minimum separation $R$, where we set $N = 100, B = 1$. The code for all experiments can be found through https://github.com/Starrskyy/debias_fl. Figure 5 shows the total variation distance of the evolution of client sampling distributions to their corresponding stationary $\pi_R$'s. Clearly increasing $R$, the convergence rate of "effective" client sampling distribution to the stationary distribution also increases, implying the decrease of mixing time $\tau_{mix}$ (see Appendix B for details). Combining this observation together with Theorems 1 and 3 leads to that larger $R$ implies faster convergence rate, which then consistently explains the observation in Figure 3. However, the above explanation is only from an empirical perspective. More rigorous explanations need theoretical advance in the convergence results to reveal explicitly the relation between the rates and values of $R$.

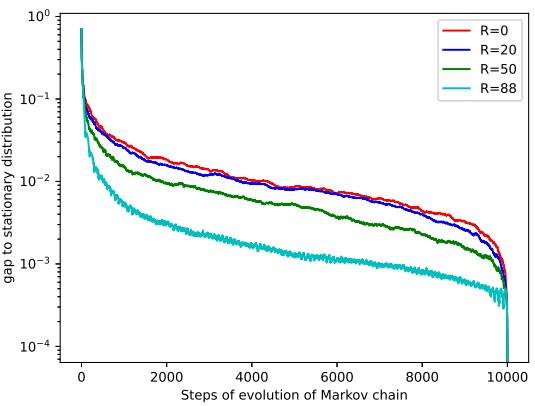

Figure 5: Convergence of client sampling distribution to $\pi_R$ for different $R$ ($N = 100, B = 1$).

