# OpenReview forum: "Debiasing Federated Learning with Correlated Client Participation"
_ICLR.cc/2025/Conference — ICLR 2025 Poster_

### Official Review · Reviewer_ehNB · 2024-10-26

**Soundness:** 2
**Presentation:** 2
**Contribution:** 2
**Rating:** 5
**Confidence:** 3

**Summary:**

Existing federated learning algorithms assume clients are sampled uniformly at random at each iteration which does not reflect the real scenario. In this paper, the authors assume that each client requires a minimum separation of R rounds between sampling. Then they model client selection as a Markov chain to theoretically analyze the setting and propose a debiasing algorithm with provable guarantees.

**Strengths:**

- Motivation: the minimum separation in federated learning is reasonable to analyze.
- Literature review is thorough.

**Weaknesses:**

- The FL setting is not rigorous. In line 121, authors use $p_i$ to capture the willingness to be sampled at each iteration which means a client may not join the training for arbitrarily long amount of iterations (with small probability). However, in line 128~129, it is claimed that "the cyclic participation corresponds to the case, R = N / B − 1", which means in the last round of the cycle, all of the remaining B clients will definitely be sampled. So the setting is not consistent. Besides, forcing clients to join cross-device federated learning is not practical.

- As has been mentioned in the "Limitations" section, the theoretical results do not enjoy linear scalability.

- I am not very convinced that Markov-chain Model is necessary to analyze the problem. The algorithm 1 essentially only try to estimate p_i and then inversely scale 1/pi to gradients in order to have uniform weight to all clients.

- The presentation of the paper can be improved.

**Questions:**

Are Markov-chain model really necessary to analyze this problem?

---

> ### Author Response · Authors · 2024-11-19
> **Response to Reviewer ehNB**
>
> **Response to Weakness 1:**
> There seems to be a misunderstanding of our problem setting and therefore its applicability and associated contribution. We clarify our setup here. Our participation pattern is as following:
> 1. Each client $i$ is sampled with probability proportional to $p_i$, **only after** it has waited for at least $R$ rounds;
>
> 2. Otherwise, client $i$ is not sampled.
>
> 3. At every round, $B$ clients are sampled to participate in the system.
>
> Thus, as cyclic participation in literature (see Cho et al., 2023 in References), sampling $B$ clients every round forces them to participate exactly once within every $N/B$ rounds. Moreover, we note that our setting is much more general compared to literature, although there is still a gap to the real-world scenario. In particular, setting $R=0$, we reduce to uniformly and independently sampling of clients; setting $R=1$, we reduce to conventional first-order Markov chain case; and setting $R=M-1$, we recover the cyclic participation case. More importantly, our introduced markov-chain framework provides a systematic way to quantify the bias under non-uniform and time-dependent participation, and then allow us to propose solutions to resolve the bias. This is the main contribution and novelty of this paper, which is totally not captured by literature.
>
> **Response to Weakness 2:**
> We acknowledge that our convergence bounds do not enjoy speedup as in literature. This is actually due to the technical difficulty in dealing with time-correlated client sampling. We note that all literature enjoying speedup in convergence relies on the assumption of independent client sampling, which makes the analysis much tractable compared to ours. The only relevant setting that theoretically studies time-correlated participation is (Cho et al., 2023), where it forces clients to participate exactly once within every $M=N/B$ rounds (i.e., in a cyclic way) which is quite restrictive compared to ours. Even in the cyclic participation case, the convergence bounds shown in (Cho et al., 2023) grow with respect to $M$, while ours get rid of the dependence on $M$. How to get a speedup in the convergence analysis is interesting and important and we hope to address it in the future.
>
> **Response to Weakness 3:**
> We believe that the Markov-chain modeling is an effective way to solve this problem. Actually, we have to point out that Algorithm 1 does not try to estimate $p_i$, but rather it tries to estimate $\pi_i$ which is the stationary distribution induced by the Markov chain. The values of $p_i$ define the transition probability matrix (see eqs. (3)-(5) for details) of the Markov chain, which has a stationary distribution $\pi$, i.e., its left eigenvector corresponding to eigenvalue 1. But the values of $p_i$ do not directly translate to values of $\pi_i$.  There is a fundamental difference between $p_i$ and $\pi_i$. Due to time correlation, the “effective” probability of client $i$ to be sampled at round $t$ is no longer $p_i$. Instead, the “effective” sampling distribution is now a time-varying distribution defined in eq. (5). If $R=0$, then it reduces to $p_i$ because now clients are sampled independently. But this is generally not true when $R > 0$! We also recommend the reviewer to refer to Lines 1340-1343 in the proof of Proposition 3 in Appendix F, where $p_1 = p_2 = 0.25, p_3=0.5$ but $\pi_1=\pi_2=0.3, \pi_4=0.4$, indicating that $\pi_i$ and $p_i$ are not the same.
>
> Therefore, although Algorithm 1 counts the participation times of clients to estimate $\pi$ which seems simple, this estimator is not independent but time-correlated. We recommend the reviewer to refer to Theorem 5 and Corollary 2 in Appendix G to see that the convergence of the estimator involves much more careful analysis and technical difficulties.
>
> **Response to Weakness 4:**
> We kindly request the reviewer to elaborate their feedback on the presentation, and would be happy to discuss and adopt concrete suggestions. Meanwhile, we updated the manuscript to include more explicit definitions of some of the notations which are marked in red.
>
> **Response to Question:**
> There might be some other techniques to analyze this problem, but the Markov-chain framework offers an effective way to systematically solve it. Leveraging such Markov-chain framework, we are able to characterize the bias suffered by existing FL algorithms and then propose methods to mitigate the bias. Moreover, note that there is no existing analysis for such a practical problem in literature and we are the first to provide theoretical characterization.

---

> > ### Comment · Reviewer_ehNB · 2024-11-26
> > **Response**
> >
> > I acknowledge that I have read the author response and decided the give higher scores.

---

> > > ### Author Response · Authors · 2024-11-28
> > > **Further request on the discussion**
> > >
> > > Dear Reviewer ehNB,
> > >
> > > We appreciate your response and your time reading our rebuttal. We also noticed that you updated your score from 3 to 5 after our rebuttal, which we are pretty grateful for. However, we were wondering if there are any specific remaining concerns or areas for improvement that prevented you from increasing your score further (to a positive one).
> > >
> > > If you could share any additional thoughts on how we might address your further concerns and make the paper stronger, it would be greatly appreciated. We look forward to hearing your thoughts.
> > >
> > > Best regards,
> > >
> > > Authors of paper 7365

---

### Official Review · Reviewer_Cr1N · 2024-10-28

**Soundness:** 3
**Presentation:** 2
**Contribution:** 3
**Rating:** 6
**Confidence:** 3

**Summary:**

This paper finds that traditional FL algorithms like FedAvg assume clients participate independently and uniformly, which is unrealistic in practical applications. It addresses the bias in FL due to non-uniform and time-correlated client participation. The authors introduce a Markov chain model to simulate the sequential and dependent nature of client participation, where each client waits a minimum number of rounds before participating again. A debiasing algorithm for FedAvg is proposed to improve convergence and ensure unbiased model updates. Empirical results also demonstrate that Debiasing FedAvg effectively reduces bias during training.

**Strengths:**

1. The authors find common FL assumption that clients participate independently and uniformly is unrealistic.
2. The paper frames client participation as a Markov process, capturing real-world constraints and interdependencies among clients.
3. The paper proposes Debiasing FedAvg converging to an unbiased solution with theoretical analysis.
4. Experiments on both synthetic and real datasets validate the algorithm’s effectiveness.

**Weaknesses:**

1. The paper claims that a larger minimum separation $R$ reduces bias. However, it lacks a discussion of how $R$ affects the server's model performance on the test set empirically and how to choose the best $R$.
2. The paper assumes a uniform minimum separation for all clients, which may not reflect real-world situations.

**Questions:**

1. If there is extreme heterogeneity among clients, how might a larger minimum separation $R$ impact the model's performance?

---

> ### Author Response · Authors · 2024-11-19
> **Response to Reviewer Cr1N**
>
> **Response to Weakness 1:**
> We have added the results of test accuracies among FedAvg, Debiasing FedAvg (ours) and FedVARP under Cifar10 in Appendix H in the updated manuscript. The experimental results show that increasing $R$ reduces bias and increases test accuracy for both FedAvg and FedVARP; and our Debiasing FedAvg outperforms FedAvg and FedVARP in both training and test settings (see Table 1 in Appendix H). We also highlight that the main problem studied in this paper is the bias effect induced by correlated client participation (justified by Theorem 1 and 2) and how to reduce it (justified by Theorem 3). Thus, we mainly focus on the training stage, i.e., solving problem (1) in the federated setting. Shown by our theories and experiments, the training error of server’s model output by conventional FL algorithms does suffer from unavoidable bias. And we can effectively reduce the bias, i.e., getting a smaller training error, by increasing $R$, and particularly no bias when $R=M-1$.
>
> **Response to Weakness 2:**
> We argue that a uniform $R$ is actually considered in the real-world setting. In particular, as discussed in (Xu et al., 2023) in References, a uniform minimum separation $R$ is adopted in Google Gboard to achieve privacy guarantees with larger $R$ leading to stronger privacy (Kairous et al., 2021). We still note that compared to literature where clients are sampled independently or from a cyclic pattern, our problem is much more general and captures them as special cases. Moreover, in the last paragraph of Section 5 (see Line 435), we discussed the possibility to extend uniform $R$ to client-specific $R_i$. In fact, for client-specific $R_i$, our theories (Theorems 1 and 3) still hold.
>
> **Response to Question 1:**
> We omitted the dependence on heterogeneity in the main text. Actually, our bounds are  proportional to the heterogeneity level $G^2$ (defined in Assumption 1). In Appendices G and F, one can see explicitly the effect of heterogeneity on the convergence (see Lines 1384 and 1547 for details, respectively). In particular, as heterogeneity grows, the larger the gap is between what the vanilla FedAvg converges to and the original optimal solution. With larger minimum separation, this gap shrinks. Under our debiasing algorithm, however, even with large heterogeneity, we can recover the original optimal solution.

---

> > ### Comment · Reviewer_Cr1N · 2024-11-25
> >
> > Thanks for the author's reply. I raise my score to 6.

---

### Official Review · Reviewer_Yi23 · 2024-11-04

**Soundness:** 3
**Presentation:** 3
**Contribution:** 3
**Rating:** 8
**Confidence:** 3

**Summary:**

This paper proposes a theoretical framework that models client participation in FL as a Markov chain, enabling the study of optimization convergence when clients exhibit non-uniform and correlated participation across rounds. The authors find that FL algorithms converge with asymptotic bias, which can be mitigated by increasing the minimum separation $R$. Additionally, they propose a debiasing algorithm for FedAvg, providing both theoretical and empirical performance guarantees for this approach.

**Strengths:**

1. The authors introduce a theoretical framework that models client participation in FL as a Markov chain, allowing the study of optimization convergence when when each client must wait at least $R$ rounds before participating again and has its own availability probability.
2. Through both theoretical and empirical results, the authors find that due to non-uniformity and time correlation effects, FL algorithms converge with asymptotic bias, which can be reduced by increasing the minimum separation $R$.
3. To achieve unbiased solutions, the authors propose a debiasing algorithm for FedAvg, with performance guarantees provided through both theoretical analysis and empirical evaluation.

**Weaknesses:**

1. The authors restrict the choices of $R$ to range from $0$ to $M-1$. However, the theoretical analysis only considers cases where $R$ ranges from $0$ to $M-2$. It would be beneficial to include the results for $R=M-1$.
2. In the experiments, the authors simplify the algorithm by partitioning the $N$ clients into $M$ groups, with exactly one group selected in each round. This setup does not align with the more complex proposed algorithm and is insufficient for a comprehensive evaluation of its performance.
3. The experiments are only conducted on synthetic dataset and MNIST dataset, which is relatively simple. More complex datasets (e.g., CIFAR-100, Shakespeare) and tasks (e.g., NLP) are recommended for a more comprehensive evaluation of the proposed algorithm's performance.


Minor: In line 109, delete “some”.

**Questions:**

1. Theorem 2 holds only under specific requirements. What about more general settings that relax these requirements?
2. The authors claim that each client can maintain its own specific $R_i$. In this more general setting, Theorems 1 and 3 hold without modification, while Theorem 2 becomes more challenging. What modifications would be needed to obtain Theorem 2?

---

> ### Author Response · Authors · 2024-11-19
> **Response to Reviewer Yi23**
>
> **Response to Weakness 1:**
> We acknowledge that Theorems 1 and 3 hold for $R \le M-2$. The reason is that when $R \le M-2$ the underlying Markov chain (5) is aperiodic as shown in Lemma 1, then guaranteeing the unique stationary distribution $\pi_R$ exists, based on which our analysis can go through. For $R=M-1$ since now the Markov chain is periodic, then its stationary distribution is not well-defined (so is the mixing time $\tau_{mix}$). Rather, we define $\pi_{M-1}$ to be the induced Perron vector (see Line 244). Actually, the analysis of $R=M-1$ is much easier compared to $R \le M-2$, since now the clients participate cyclically and hence it is equivalent to that clients are sampled uniformly. In particular, the term $e_2$ in the proof of Lemma 10 is zero for any $t$ and $\tau$. Therefore, as we stated in Lines 423-427, much better convergence bounds can be obtained due to such nice cyclic pattern (see Cho et al., 2023 for more details in References in the paper). The reason we did not include $R=M-1$ to our Theorems 1 and 3 is that the mixing time $\tau_{mix}$ is not well-defined in this case, since the Markov chain is periodic.
>
> **Response to Weakness 2:**
> In Algorithm 1 and all our analysis, there is no restriction to partition clients into separate “groups”. We note that our analysis holds true even for $B=1$, which would effectively remove the partition. Initially, any subset of clients with size $B$ can be sampled. Then due to the requirement of minimum separation, the group of clients becomes available at the same time. We therefore group them in experiments only for implementation simplicity. Our algorithm can still effectively reduce bias without partitioning.
>
> **Response to Weakness 3:**
> We thank the reviewer for the suggestion. We have added the experiment for Cifar 10 in the appendix. Moreover, we note that the main contribution of the paper is to theoretically analyze FL under correlated client participation. Thus we acknowledge that the experimental part of the paper might not be comprehensive enough, while we note that our experimental results effectively verify our theoretical claims.
>
> **Response to Question 1:**
> In Theorem 2, the only assumption we made is that the availability probabilities $p_i$’s are not “too far away” (characterized by $\delta$) from the uniform distribution. The reason for this assumption is due to the perturbation technique we used in the proof, which only holds for a small neighborhood around uniform distribution. We believe that some new proof technique is needed in order to remove the assumption and we hope to address it in the future.
>
> **Response to Question 2:**
> The nice thing about a uniform $R$ across clients is that the indices of clients within every $R+1$ rounds are non-repeated, which enables us to analytically get the expression of the column sum $b_R$ of $P_R$ in Appendix D.1. Then, our proof idea relies on studying the monotonicity of $b_R$. However, if each client maintains its own $R_i$, taking $R = \max_i R_i$ no longer guarantees non-repeated indices within $R+1$ rounds, which renders much technical difficulty to analyze the monotonicity of $b_R$ in the sense that the analytical expression of $b_R$ is stochastic and unknow. Therefore, our proof fails in this case. We hope to solve this problem in our future work.

---

> > ### Author Response · Authors · 2024-11-28
> > **Request for rebuttal discussion**
> >
> > Dear Reviewer Yi23,
> >
> > Thank you once again for reviewing our submission, and for taking the time to provide feedback. As we are currently in the rebuttal phase, we wanted to reach out to ensure we have fully addressed your concerns. If there are any remaining issues or aspects of the work that you believe need further clarification or improvement, we would be happy to engage in a discussion and provide additional details.
> >
> > We are more than happy to incorporate any additional feedback to enhance the quality the paper. Please let us know if there is anything further we can address or clarify to strengthen your confidence in our work. We look forward to your response.
> >
> > Best regards,
> >
> > Authors of paper 7365

---

> > > ### Comment · Reviewer_Yi23 · 2024-12-02
> > >
> > > Thank the authors for the responses. My concerns have been addressed, and I would like to increase my score.

---

### Official Review · Reviewer_GYuM · 2024-11-11

**Soundness:** 3
**Presentation:** 2
**Contribution:** 3
**Rating:** 8
**Confidence:** 4

**Summary:**

This paper studied the federated learning problem with partial client participation. In particular, it focused on the case where there is minimum separate between clients in terms of minimum rounds. The authors formulated the client participation process as a R-th order Markov chain and characterize the marginal stationary distribution of clients to be sampled. The authors proposed a debiasing FedAvg based on the estimation of this marginal stationary distribution and provided the convergence analysis of the proposed algorithm as well as the original FedAvg algorithm. There are several very interesting observations of this paper. The performance of the proposed algorithm is also verified using simulations.

**Strengths:**

1. The authors formulates the client participation process as a R-th order Markov chain.

2. The authors proposed the debiasing FedAvg algorithm based on the estimation of marginal stationary distribution of clients to be sampled.

3. The authors provided the convergence analysis of both FedAvg (to indicate the problem) and the proposed algorithm which can converge.

**Weaknesses:**

1. The paper is not well written and there are some notations not explained, e.g., $\tau_{mix}$ (is it the mixing time?) and $p_e$, although the paper presented quite a few interesting ideas.

2. The authors discussed quite a few limitations of the proposed approach and its proofs. These seem the weaknesses of the paper.

**Questions:**

1. The algorithm is based on GD instead of SGD. If SGD was used, will there be any challenges?

2. I suggest the authors put the proof of Proposition 1 in the appendix.

**Details Of Ethics Concerns:**

N/A.

---

> ### Author Response · Authors · 2024-11-19
> **Response to Reviewer GYuM**
>
> **Response to Weakness 1:**
> We thank the reviewer for pointing out the presentation issue. We defined $\tau_{mix}$ as the mixing time in the statement of Theorem 1 (see Line 297)  and $p_e$ denotes the $e$-th client availability probability. We also made corresponding changes in our updated version.
>
> **Response to Weakness 2:**
> We acknowledge the weakness and we provided discussions in Section 7. As stated in Section 7, the limitations are technical difficulties of our analysis. This paper serves as a first step towards analysis of federated learning under correlated client participation. We would like to look for more advanced mathematical tools to solve them in the future.
>
> **Response to Question 1:**
> In our analysis, we just applied local gradient descent in the local updates for simplicity. And our analysis can be easily extended to local SGD. In such case, we can similarly bound $\Vert x_{t+1} - x_t \Vert^2$ as in Lemma 7 by introducing bounded variance of stochastic gradient for each client (which is a standard assumption in stochastic optimization analysis). Then, all the following analysis goes through as well, while the final convergence bounds additionally depend on the variance of the stochastic gradient. Moreover, we highlight that in the experiment, we did use local SGD during local updates, and the results verify our theoretical demonstration.
>
> **Response to Question 2:**
> We thank the reviewer for the suggestion. We have added the proof of Proposition 1 in our updated Appendix C for clarity.

---

### Author Response · Authors · 2024-11-19
**General response to all reviewers**

We appreciate all valuable comments and suggestions provided by the reviewers. We have fully addressed all the concerns and made corresponding editions (in red) in the updated manuscript. We are eager to discuss with the reviewers for any further comment.

---

### Meta-Review · Area_Chair_Aovg · 2024-12-15

**Metareview:**

This introduces a theoretical framework modeling client participation in federated learning (FL) as a Markov chain, addressing non-uniform and correlated participation across rounds. The authors analyze scenarios where clients must wait a minimum number of rounds before re-participating, demonstrating that increasing this minimum separation reduces bias caused by non-uniform client availability. They also propose a debiasing algorithm for Federated Averaging (FedAvg) that converges to the unbiased optimal solution under arbitrary minimum separation and unknown client availability distributions.

Reviewers acknowledged the novelty of modeling client participation as a Markov chain and appreciated the practical relevance of addressing correlated client participation in FL systems. The proposed debiasing algorithm was noted as a significant contribution to improving convergence in realistic FL scenarios. Reviewers found the mathematical rigor appropriate and the assumptions reasonable for the FL context. The empirical results supporting the theoretical claims were well-received. However, some reviewers suggested that additional experiments on diverse datasets and with varying system parameters could strengthen the validation of the proposed methods.

In response to the feedback, the authors provided clarifications on the Markov chain modeling approach and supplied additional experimental results addressing concerns about empirical validation. They also revised the manuscript to improve clarity, especially in sections detailing the theoretical framework and algorithmic procedures.

---
Considering the reviewers' assessments and the authors' responses, the decision is to accept the paper. The work presents a novel and practically significant approach to addressing correlated client participation in federated learning through Markov chain modeling and a debiasing algorithm. The theoretical contributions are well-substantiated, and the empirical validations, though with room for further expansion, support the claims made. The authors' revisions have adequately addressed the primary concerns regarding clarity and empirical evidence.

**Additional Comments On Reviewer Discussion:**

All the reviewers recommend accept or increased their score during the rebuttal period.

---

### Decision · Program_Chairs · 2025-01-22

Accept (Poster)